# The membrane periodic skeleton is an actomyosin network that regulates axonal diameter and conduction

Ana Rita Costa[1,2], Sara C Sousa[1,2,3], Rita Pinto-Costa[1,2], José C Mateus[2,3,4], Cátia DF Lopes[2,4], Ana Catarina Costa[1,2,3,4], David Rosa[1,2], Diana Machado[1,2], Luis Pajuelo[1,2], Xuewei Wang[5], Feng-quan Zhou[5], António J Pereira[2,6], Paula Sampaio[2,7], Boris Y Rubinstein[8], Inês Mendes Pinto[9], Marko Lampe[10], Paulo Aguiar[2,4], Monica M Sousa[1,2]*

[1]Nerve Regeneration Group, Porto, Portugal; [2]i3S- Instituto de Investigação e Inovação em Saúde, Universidade do Porto, Porto, Portugal; [3]ICBAS- Instituto de Ciências Biomédicas Abel Salazar, Universidade do Porto, Porto, Portugal; [4]Neuroengineering and Computational Neuroscience Group, INEB- Instituto de Engenharia Biomédica, Universidade do Porto, Porto, Portugal; [5]Department of Orthopaedic Surgery, Johns Hopkins University School of Medicine; The Solomon H. Snyder Department of Neuroscience, Johns Hopkins University School of Medicine, Baltimore, United States; [6]Chromosome Instability and Dynamics Group, Porto, Portugal; [7]Advanced Light Microscopy, IBMC- Instituto de Biologia Molecular e Celular, Universidade do Porto, Porto, Portugal; [8]Stowers Institute for Medical Research, Kansas City, United States; [9]International Iberian Nanotechnology Laboratory, Braga, Portugal; [10]Advanced Light Microscopy Facility, EMBL, Heidelberg, Germany

*For correspondence:
msousa@ibmc.up.pt

Competing interests: The authors declare that no competing interests exist.

**Abstract** Neurons have a membrane periodic skeleton (MPS) composed of actin rings interconnected by spectrin. Here, combining chemical and genetic gain- and loss-of-function assays, we show that in rat hippocampal neurons the MPS is an actomyosin network that controls axonal expansion and contraction. Using super-resolution microscopy, we analyzed the localization of axonal non-muscle myosin II (NMII). We show that active NMII light chains are colocalized with actin rings and organized in a circular periodic manner throughout the axon shaft. In contrast, NMII heavy chains are mostly positioned along the longitudinal axonal axis, being able to crosslink adjacent rings. NMII filaments can play contractile or scaffolding roles determined by their position relative to actin rings and activation state. We also show that MPS destabilization through NMII inactivation affects axonal electrophysiology, increasing action potential conduction velocity. In summary, our findings open new perspectives on axon diameter regulation, with important implications in neuronal biology.

## Introduction

When considering an adult axon, its diameter can oscillate depending on organelle transport (*Greenberg et al., 1990*), neuronal activity (*Fields, 2011*), deformations generated by movement or degeneration. The mechanisms controlling axonal diameter throughout the neuronal lifetime remain however unclear. The mature axon shaft is supported by a submembraneous actin-spectrin network- the membrane periodic skeleton (MPS)- composed of actin rings regularly spaced by spectrin tetramers approximately every 190 nm (*Xu et al., 2013*). Although its assembly and function are largely

unknown, the MPS may provide mechanical support for the long thin structure of axons (*Hammarlund et al., 2007*). In the initial MPS model, each ring was hypothesized to be composed of actin filaments capped by the actin-binding protein adducin (*Xu et al., 2013*). Recently, combining platinum-replica electron and optical super-resolution microscopy, the MPS actin rings were shown to be made of two long, intertwined actin filaments (*Vassilopoulos et al., 2019*). According to this novel view, adducin might be responsible to enhance the lateral binding of spectrin to actin. We have previously demonstrated that adducin is required to maintain axon caliber as its absence in vitro leads to actin rings of increased diameter, while in vivo it results in progressive axon enlargement and degeneration (*Leite et al., 2016*). We have additionally found that in vitro, the radius of axonal actin ring narrows over time (*Leite et al., 2016*), supporting that the MPS has dynamic properties. Since reduction in axon diameter with time occurs both in WT and α-adducin knock-out (KO) neurons, MPS dynamics is probably regulated by additional actin-binding proteins.

The role of actin in the control of axonal radial tension is emerging (*Costa et al., 2018*; *Fan et al., 2017*). NMII is a hexamer composed by two heavy chains, two regulatory light chains (RLC) and two essential light chains (ELC), being a conserved molecule for generating mechanical forces (*Vicente-Manzanares et al., 2009*). The NMII contractile ATPase activity and the assembly of myosin filaments that coordinate force generation is activated by phosphorylation of myosin light chain (MLC) (*Vicente-Manzanares et al., 2009*). Here, we provide evidence that the axonal MPS, similarly to actin rings present in other biological contexts, is an actomyosin-II network that regulates circumferential axonal contractility. Furthermore, we demonstrate that the MPS affects signal propagation velocity, a property with important functional implications.

## Results and discussion

### Modulation of NMII activity regulates the expansion and contraction of axonal diameter

The MPS of both WT and α-adducin KO neurons contracts in vitro at a rate of 6–12 nm/day (*Leite et al., 2016*). Given the general role of NMII in promoting contractility, we tested whether axon thinning in vitro was dependent on NMII activity. For that, NMII-mediated ATP hydrolysis and thereby actomyosin-based motility, were inhibited by blebbistatin (*Straight et al., 2003*; *Figure 1A*). In the presence of the drug, axon thinning of hippocampal neurons from DIV8 to DIV22 was abolished as determined using Stimulated Emission Depletion (STED) microscopy (*Figure 1B,C*). This supports that axon thinning in vitro occurs through a NMII-mediated mechanism. Additionally, DIV8 hippocampal neurons treated with blebbistatin had a 1.3-fold increase in axon diameter (*Figure 1D,E*). Alternative modes of drug-mediated modulation of myosin activity were tested, including ML-7 (*Saitoh et al., 1987*), calyculin A (*Ishihara et al., 1989*), and myovin1 (*Gramlich and Klyachko, 2017*; *Islam et al., 2010*). The function of NMII is controlled by MLC kinase (MLCK) that phosphorylates the NMII RLCs leading to conformational changes and self-assembly in myosin filaments (*Vicente-Manzanares et al., 2009*; *Figure 1A*). ML-7, a selective MLCK inhibitor that decreases pMLC levels in hippocampal neurons (*Figure 1—figure supplement 1A, B*), led to an increase in axonal diameter similar to that produced by blebbistatin (*Figure 1D,E*). As protein phosphatase 1 (PP1) is the major myosin phosphatase responsible for dephosphorylation of NMII (*Matsumura and Hartshorne, 2008*), calyculin A, a potent PP1 inhibitor that results in increased phosphorylation of NMII RLCs and increased myosin contractility, was used (*Iizuka et al., 1999*; *Figure 1A*). In the presence of calyculin A, sustained activation of NMII resulted in a decrease to 0.8-fold in axonal diameter (*Figure 1D,E*), supporting that NMII activity can positively and negatively regulate axonal caliber. Of note, the effect of calyculin A was reverted by blebbistatin (*Figure 1D,E*). Myovin-1, a potent inhibitor of the ATPase activity of myosin V that inhibits myosin V-dependent intersynaptic vesicle exchange in hippocampal neurons (*Gramlich and Klyachko, 2017*), had no effect on the MPS diameter (*Figure 1D,E*) suggesting a NMII-specific effect. In further support of the MPS dynamic nature, the effect of blebbistatin was reverted as drug treatment was released from DIV8 to DIV12 (*Figure 1F,G*). In summary, we show that axon diameter can be controlled both at the level of NMII activation through phosphorylation, and at the level of its power-stroke activity.

We next asked whether reducing both adducin levels and NMII activity might have a cumulative effect. To test this hypothesis, the MPS was examined in blebbistatin-treated hippocampal neurons

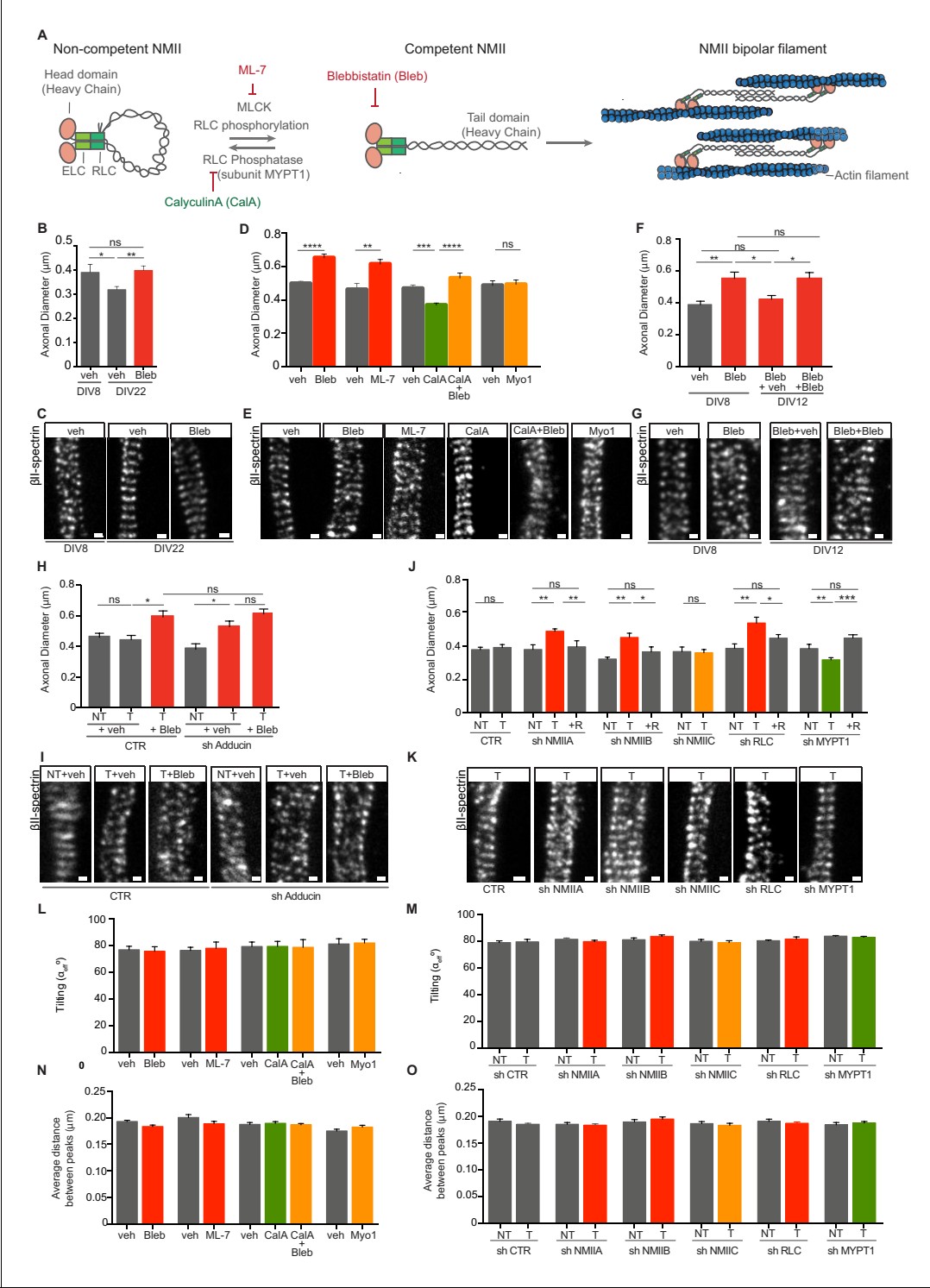

**Figure 1.** Modulation of NMII activity regulates axon diameter. (**A**) Representation of NMII structure and activity regulation. NMII contains two heavy chains (HC), two RLCs and two ELCs. The head domain includes an actin-binding site and an ATPase motor domain. ELCs and RLCs are bound to NMII head domains. In the absence of RLC phosphorylation, NMII is in a non-competent conformation. Upon RLC phosphorylation by MLCK, NMII unfolds to generate a competent form. NMII is then able to assemble into bipolar filaments, which bind to actin. MLC phosphatase that includes a regulatory subunit -MYPT1- can revert this activation. NMII activity can be modulated by drugs including ML-7, a specific MLCK inhibitor; calyculin A (CalA) an inhibitor of RLC phosphatase, and blebbistatin (Bleb) that inhibits NMII-mediated ATP hydrolysis. (**B**) Axon diameter of DIV8 and DIV22 hippocampal neurons incubated either with vehicle (veh) or 3 μM blebbistatin (Bleb) at DIV3 and DIV8 (1 hr before fixation); n = 6–8 axons/condition and 20–87 rings/axon. (**C**) Representative STED images of βII-spectrin immunostaining using a STAR 635P secondary antibody related to (**B**). (**D**) Axon diameter of DIV8

*Figure 1 continued on next page*

*Figure 1 continued*

rat hippocampal neurons incubated with drug modulators of myosin activity: 3 μM blebbistatin (Bleb), 4 μM ML-7, 5 nM calyculin A (CalA), 5 nM calyculin A + 3 μM blebbistation (CalA+Bleb), 4 μM myovin1 (Myo1) and the respective vehicles (veh). Calyculin A was added at DIV8 (25 min before fixation), while the remaining drugs were added at DIV3 and DIV8 1 hr before fixation; n = 8–13 axons/condition and 10–77 rings/axon. (E) Representative STED images of βII-spectrin immunostaining using a STAR 635P secondary antibody related to (D). (F) Axon diameter of DIV8 rat hippocampal neurons incubated with either 3 μM blebbistatin (Bleb) or vehicle (veh) at DIV3 and DIV8 (1 hr before fixation). In cells treated with blebbistatin up to DIV8, drug was replaced by either vehicle (Bleb+veh) or drug treatment was continued (Bleb+Bleb) and axonal diameter was analysed at DIV12; n = 7–10 axons/condition and 12–79 rings/axon. (G) Representative STED images of βII-spectrin immunostaining using a STAR 635P secondary antibody related to (F). (H) Axon diameter of DIV8 rat hippocampal neurons upon shRNA-mediated knockdown of adducin (sh Adducin) or transfection using a control shRNA (CTR) and subsequent incubation with either Bleb or veh; T- transfected; NT- non-transfected; n = 4–13 axons/ condition and n = 12–88 rings/axon. (I) Representative STED images of βII-spectrin immunostaining using a STAR 635P secondary antibody related to (H). (J) Axon diameter of DIV8 rat hippocampal neurons upon shRNA-mediated knockdown of different NMII isoforms (NMIIA, NMIIB and NMIIC), RLC or MYPT1; scramble shRNA was used as control (CTR); shRNA resistant constructs (R) for each specific shRNA were used to rescue their effect; T- transfected; NT- non-transfected; n = 5–15 axons/condition and 6–103 rings/axon. (K) Representative STED images of βII-spectrin immunostaining using a STAR 635P secondary antibody related to (J). (L) Tilting (effective angle α$_{eff}$) of the actin rings in relation to the axonal axis after incubation with NMII chemical modulators; similar concentrations and incubation times as defined in (D); n = 8–14 axons/condition and 14–72 rings/axon and (M) after shRNA-mediated downregulation; n = 5–15 axons/condition and 7–70 rings/axon. Tilting was quantified in STED images after βII-spectrin immunostaining. (N) MPS periodicity after incubation with myosin-targeting drugs; similar concentrations and incubation times as defined in (D). The average distance between peaks was quantified by STED after βII-spectrin immunostaining; n = 8–14 axons/condition and 14–72 rings/axon. (O) MPS periodicity after shRNA-mediated knockdown of different NMII isoforms (NMIIA, NMIIB and NMIIC), RLC or MYPT1; a scramble shRNA was used as control (CTR). The average distance between peaks was quantified by STED after βII-spectrin immunostaining; T- transfected; NT- non-transfected; n = 5–15 axons/condition and 7–70 rings/axon. In all panels: Scale bars, 200 nm. *p<0.05; **p<0.01; ***p<0.001; ****p<0.0001; ns-non significant. Graphs show mean ± s.e.m. In every case displayed in this Figure, at least 3 independent experiments have been performed.

The online version of this article includes the following figure supplement(s) for figure 1:

**Figure supplement 1.** Analysis of ML-7 activity and ShRNA-mediated downregulation.

where adducin was knocked-down (*Figure 1H,I* and *Figure 1—figure supplement 1C, D*). Whereas inhibiting NMII activity or knocking down adducin led to a 1.3-fold increase in axonal caliber, being the latter case in agreement with our previous studies in adducin KO neurons (*Leite et al., 2016*), the combination of both treatments did not result in a cumulative effect (*Figure 1H,I*).

In mammalian cells, different genes encode three different NMII heavy chain isoforms, NMHCIIA, NMHCIIB and NMHCIIC that determine the NMII isoform (NMIIA, NMIIB and NMIIC, respectively) (*Vicente-Manzanares et al., 2009*). Different NMII isoforms have unique kinetic properties and specific cellular functions (*Vicente-Manzanares et al., 2009*). To understand the role of different NMII isoforms in the MPS, we induced their down-regulation using specific shRNAs (*Figure 1J,K* and *Figure 1—figure supplement 1C-L*). Downregulation of either NMIIA or NMIIB induced a 1.3-fold increase in axon caliber, similar to that induced by either blebbistatin or ML-7 treatment, whereas downregulation of NMIIC was without effect (*Figure 1J,K*). These results are in agreement with previous studies highlighting the distinctive mechanochemical profile of NMIIC and lower ability in generating contractile force when compared with NMIIA and NMIIB isoforms (*Billington et al., 2013*). As such, knockdown of either NMIIA or NMIIB was sufficient to induce a comparable outcome as drug-mediated inhibition of all NMII isoforms, whereas NMIIC probably does not participate in NMII-mediated regulation of axon diameter. shRNA-mediated knockdown of NMII RLC (*Wang et al., 2008*) had a similar effect to the knockdown of myosin heavy chains, leading to MPS expansion (*Figure 1J,K*). Myosin phosphatase target subunit 1 (MYPT1) is the regulatory subunit of myosin light chain phosphatase and is responsible for the dephosphorylation of MLC. The inactivation of its enzymatic activity increases NMII phosphorylation and consequently its activation (*Vicente-Manzanares et al., 2009*; *Figure 1A*). Knocking down MYPT1 (*Figure 1* and *Figure 1—figure supplement 1E, F, K, L*) caused a decrease to 0.8-fold in axon diameter (*Figure 1J,K*). Of note, rescue experiments performed using co-transfection with shRNA-resistant constructs reverted the phenotype of the corresponding shRNA, confirming their specificity (*Figure 1J,K*). Our data support that both NMIIA and NMIIB, contribute to the MPS actomyosin network.

The possible tilting of actin rings in relation to the axonal axis was measured in all the conditions where drug- or shRNA-mediated modulation of NMII activity was performed. No variations in the angle of actin rings in relation to the axonal axis were found (*Figure 1L,M*) suggesting that NMII probably does not contract between adjacent rings. Of note, tilting may be hindered by the

existence of the βII-spectrin scaffold. Also, having several active NMII filaments attached to adjacent axon rings, may generate a net force that would be averaged out and thus close to zero. Importantly, neither drug- nor shRNA-mediated modulation of NMII activity, resulted in alterations of MPS periodicity (*Figure 1N,O*, respectively), thus arguing that a possible participation of NMII in axonal longitudinal tension does not interfere with the 190 nm length of extended spectrin tetramers (*Brown et al., 2015*).

## Propagation velocity is altered by manipulation of NMII activity

Neuronal electrophysiology is intimately related to morphology. Besides the relation between conduction velocity and axonal caliber (*Hodgkin, 1954*; *Waxman, 1980*), sodium channels are distributed in axons in a periodic pattern coordinated with the MPS (*Xu et al., 2013*). To uncover possible electrophysiological implications of manipulating NMII activity, we used a combination of in vitro microelectrode arrays (MEA) and microfluidics (µEF platforms) to characterize axonal propagation velocity between DIV11 and DIV14. This µEF platform was previously shown by us to allow axon alignment on top of a sequence of microelectrodes, providing detailed electrophysiological information about signal conduction on isolated axons (*Figure 2A*) while also recording global activity levels (*Heiney et al., 2019*; *Lopes et al., 2018*). Blebbistatin-treated and untreated neurons were electrophysiologically competent regarding signal conduction and no significant differences in global electrical activity were found, as determined by assessing their mean firing rate (MFR) (*Figure 2B,C*). However, axonal conduction velocity between treated and untreated axons was different, with distinct velocity distributions (*Figure 2D,E*). From DIV11 on, blebbistatin-treated neurons consistently showed higher propagation velocities when compared to vehicle-treated cells. In unmyelinated axons, velocity conduction is proportional to the square root of the axonal diameter (*Hodgkin, 1954*) and blebbistatin treatment produces a 1.3-fold increase in axon diameter that is not altered throughout time (*Figure 1F*). Our data favors that NMII modulation affects axonal electrophysiology through changes in axonal diameter. This effect should be further explored by using additional modulators of NMII activity and shRNA-mediated downregulation approaches.

## Phosphorylated NMII light chains are colocalized with actin within the MPS and organized as circular periodic structures persisting throughout the axon shaft

We next analyzed the localization of NMII in axons using STED microscopy, and 3D Single Molecule Localization Microscopy (SMLM) with a dSTORM/GSDIM-protocol (*Fölling et al., 2008*; *Heilemann et al., 2008*). pMLC has been described to be specifically localized in the axon initial segment (AIS) (*Berger et al., 2018*). pMLC regulates the head groups of the NMII heavy chains, mediating their binding to actin and the organization of NMII into bipolar filaments (*Berger et al., 2018*). Despite the enrichment of pMLC at the AIS, not only non-phosphorylated MLC, but also active pMLC could be observed throughout the entire axon shaft (*Figure 3A*). By applying SMLM to the AIS of DIV8 hippocampal neurons and focusing on their lateral outermost points up to the axonal surface, pMLC displayed a bilateral ~200 nm periodicity and a circular structure (*Figure 3B,C*; *Video 1*). Periodic (~190 nm) pMLC staining, coincident with phalloidin staining restricted to the AIS, has been previously shown using STORM (*Berger et al., 2018*), suggesting a direct binding of pMLC to actin. Here, we show that pMLC periodicity is not restricted to the AIS, extending to the axon shaft, being visible in the lateral outer part of the axon (*Figure 3D*- highlighted by white ruler; and *3E*). Interestingly, in some axonal regions, pMLC showed a periodic distribution, consistent with anchoring in different positions of adjacent actin rings (*Figure 3D*- highlighted by red ruler; and *Figure 3F*). The absence of a generalized striped pattern for pMLC (in contrast to what is observed for axonal actin and βII-spectrin staining), supports that in each actin ring a limited number of pMLC molecules is anchored. Alternatively, technical limitations of the antibodies used may preclude the visualization of the entire pool of pMLC molecules present in the structure. However, in some regions of the axon shaft of hippocampal neurons, pMLC not only displays the expected striped pattern but also is colocalized with actin, supporting that throughout the axon, pMLC is bound to actin rings that compose the MPS (*Figure 3G*). Simultaneous analysis of βII-spectrin and pMLC using dual color SMLM showed that in the axon shaft the two molecules intercalate (*Figure 3H*, arrowheads highlight pMLC and asterisks βII-spectrin), whereas a periodic distribution of pMLC consistent with

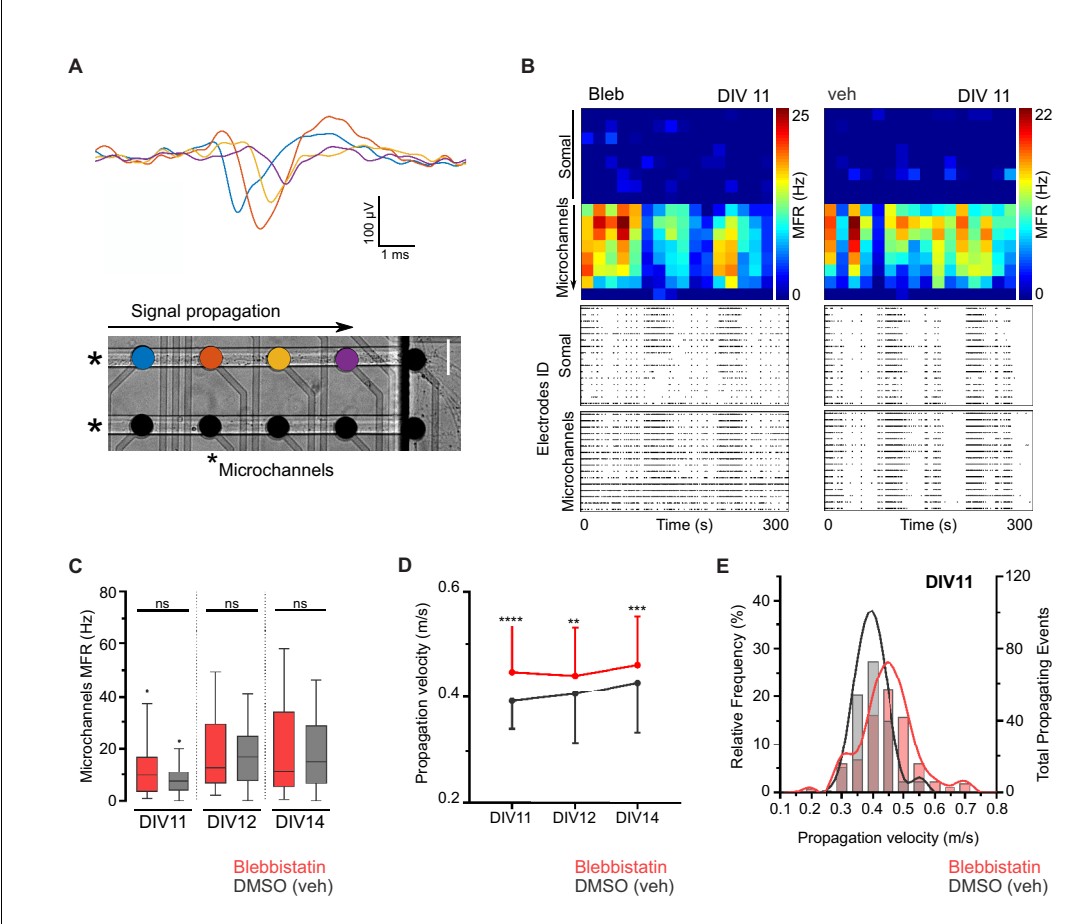

**Figure 2.** Blebbistatin treatment increases axonal signal propagation velocity. (A) Single propagating event recorded along four electrodes inside a microchannel. Each colored signal trace corresponds to an electrode from a single microchannel. Two microchannels are indicated by asterisks in the phase-contrast image of hippocampal neurons cultured on a µEF platform at DIV8 (scale bar = 50 µm). (B) Activity maps of vehicle and 3 µM blebbistatin (Bleb) treated cultures at DIV11. Each pixel corresponds to one recording electrode and the mean firing rate (MFR) is color-coded for each electrode. Notice that µEFs allow for the recording of both somal and axonal activity. Below are shown raster plots of 300 s of activity from 16 active electrodes of the somal compartment and from 16 electrodes within the microchannels (one electrode per microchannel). (C) MFR from the electrodes in the microchannels for vehicle and Bleb-treated cultures (DIV11-14). (D) Median propagation velocity (± SD) for vehicle and blebbistatin-treated cultures at DIV11, 12, and 14 (propagating events pooled from 25 to 43 microchannels, from 3 to 4 independent µEFs). Unpaired t-test or Mann-Whitney test; ns = not significant, **p<0.01, ***p<0.001 and ****p<0.0001. (E) Frequency distribution of the propagation velocity domains (0.05 m/s binning) at DIV11 for both conditions.

anchoring in different positions of adjacent actin rings also occurs, in both cases non-overlapping with βII spectrin (*Figure 3H*).

## NMII heavy chains are present along the longitudinal axonal axis, being able to crosslink adjacent rings

To characterize the spatial location of NMII heavy chains, we used antibodies that recognize the C-terminal tail region of NMIIA and NMIIB heavy chains, which label the middle portion of NMII bipolar filaments (*Figure 4A*). Using these antibodies, we started by performing two-color STED in combination with phalloidin staining. When focusing the axonal lateral outermost points, both NMII heavy chain A (*Figure 4B*) and NMII heavy chain B (*Figure 4C*) were mainly detected within the axon core. At discrete sites, NMIIA and NMIIB colocalized with phalloidin (*Figure 4B,C*- highlighted with arrowheads), suggesting that NMII heavy chains might additionally be within individual actin rings. Although no simultaneous labeling of NMIIA and NMIIB was performed given host antibody constraints, their localization within axons did not differ significantly (*Figure 4B,C*). When observing

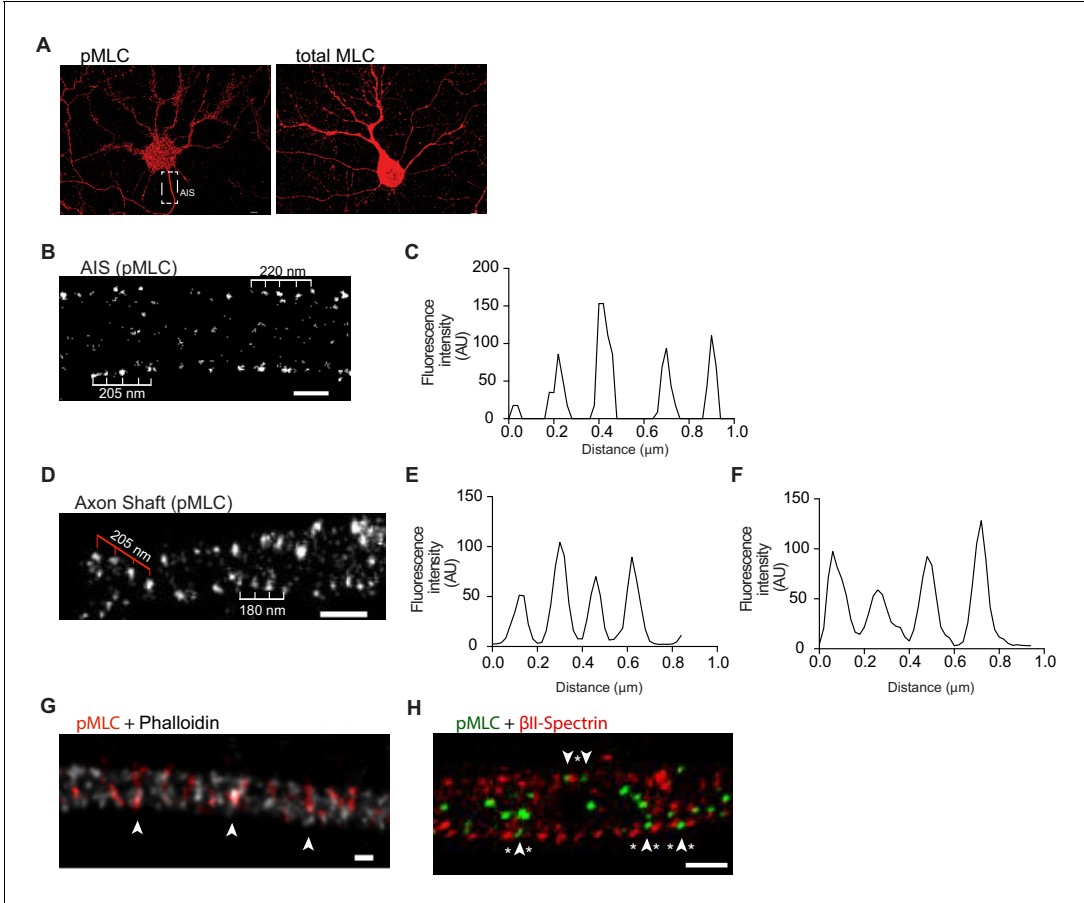

**Figure 3.** Phosphorylated NMII light chains are organized as circular periodic structures persisting throughout the axon shaft. (**A**) Immunolabeling of DIV8 hippocampal neurons with rabbit anti-pMLC (left) and rabbit anti-total MLC (right) using a secondary anti- rabbit Alexa Fluor 647 antibody. Scale bar: 5 μm. (**B**) Single colour SMLM of pMLC distribution in the AIS using a secondary Alexa Fluor 647 antibody. Scale bar: 500 nm. (**C**) Analysis of the periodicity related to (**B**). (**D**) 2D single colour SMLM of the axon shaft of a DIV8 hippocampal neuron immunostained for pMLC using a secondary Alexa Fluor 647 antibody. Scale bar: 500 nm. Periodic distribution in outer most regions of the axon shaft is highlighted with white ruler; inner distribution consistent with anchoring in different positions of adjacent actin rings is highlighted with red ruler. (**E**) Analysis of periodicity in outer most regions of the axon shaft (highlighted in white in D). (**F**) Analysis of inner periodic distribution (highlighted in red in D). (**G**) Two colour STED of a DIV8 hippocampal neuron immunostained against pMLC (red) and stained for actin (gray), using a STAR 580 secondary antibody and phalloidin 635, respectively. pMLC molecules, highlighted with arrowheads, co-localize with actin. Scale bar: 200 nm. The raw image was deconvolved using the CMLE algorithm (Huygens Professional, Scientific Volume Imaging). (**H**) Z projection of 3D two-colour SMLM of a DIV8 hippocampal neuron immunostained against βII-spectrin (red) and pMLC (green) using secondary antibodies labeled with Alexa Fluor 532 and 647, respectively. Molecules intercalating at outermost positions are highlighted using arrowheads (pMLC) and asterisks (βII-spectrin). Scale bar: 500 nm.

axons single-labeled for NMIIA using SMLM, a pattern emerged suggesting the existence of multiple NMIIA heavy chain filaments along the longitudinal axonal axis (*Figure 4D*). Of note, this organization clearly contrasted with the circular periodic distribution of pMLC (*Figure 3B*; *Video 1*).

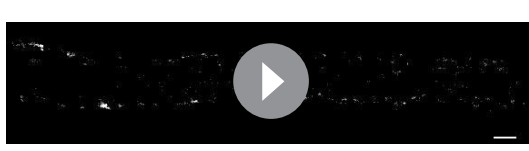

**Video 1.** 3D single colour SMLM of pMLC distribution in the AIS of a DIV8 hippocampal neuron. Scale bar: 500 nm.

https://elifesciences.org/articles/55471#video1

To further assess the localization of NMII heavy chains in axons, the SH-SY5Y neuroblastoma cell line and primary hippocampal neurons were transfected with fluorescent NMII fusion constructs that allow visualizing simultaneously the N-terminal (eGFP tag) and C-terminal (mApple tag) domains of the NMIIA heavy chain (*Beach et al., 2014*; *Figure 4E*). Of note, in differentiated SH-SY5Y cells, the approximately 190 nm MPS periodic pattern of βII-spectrin was observed (*Figure 4F,G*). We used super-

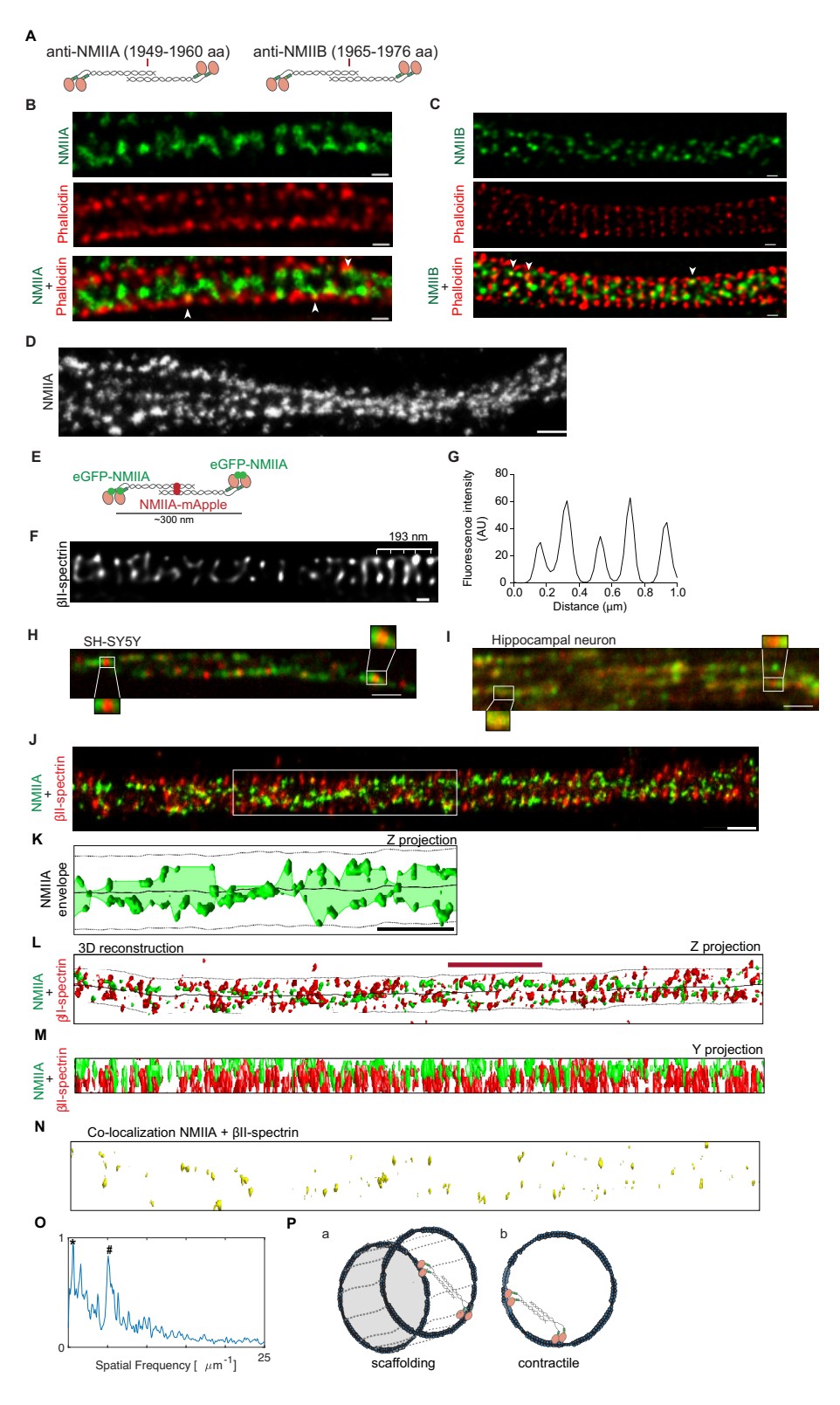

**Figure 4.** NMII heavy chains organize into filaments distributed in multiple orientations along the axon shaft. (**A**) Representation of recognition sites of antibodies against NMIIA and NMIIB (NMIIA: 1949–1960 aa; NMIIB: 1965–1976 aa). (**B**) Representative STED analysis of a DIV8 hippocampal neuron co-stained with NMIIA using a secondary anti-rabbit STAR 580 (upper) and phalloidin 635 (middle); the merged image is shown in the lower panel. Scale bar: 200 nm. (**C**) Representative STED analysis of a DIV8 hippocampal neuron co-stained with NMIIB using a secondary anti-rabbit STAR 580 (upper)

*Figure 4 continued on next page*

*Figure 4 continued*

and phalloidin 635 (middle); the merged image is shown in the lower panel. Scale bar: 200 nm. (**D**) Z projection of 3D SMLM of an axon single labeled for NMII heavy chain A using an Alexa Fluor 647 labeled secondary antibody. NMIIA labeling following different orientations in relation to the axonal axis is highlighted with arrowheads. Scale bar: 500 nm. (**E**) Representation of a bipolar NMIIA filament, with the N-terminal eGFP tagged and the C-terminal mApple tagged. (**F**) SH-SY5Y was immunolabelled with βII-spectrin using a STAR 635P secondary and imaged using STED. Scale bar: 200 nm. The raw image was deconvolved using the CMLE algorithm (Huygens Professional, SVI). (**G**) Analysis of SH-SY5Y βII-spectrin periodicity related to (**F**). (**H**) Representative spinning disk image of SH-SY5Y after co-transfection with eGFP-NMIIA and NMIIA-mApple. Scale bar: 1 µm. Insets highlight bipolar NMIIA filaments of ~300 nm. (**I**) Representative spinning disk image of a primary hippocampal neuron axon after co-transfection with eGFP-NMIIA and NMIIA-mApple. Scale bar: 1 µm. (**J**) Z projection of a 3D SMLM double stained for βII-spectrin (red) and NMII heavy chain A (green) using anti-mouse Alexa Fluor 532 and anti-rabbit Alexa Fluor 647, respectively. Scale bar: 500 nm. (**K**) Analysis of the envelope of NMIIA labeling (Z projection) relative to the region highlighted by the white box in (**J**); the axonal membrane (dashed line) and centerline (solid line) are depicted. Scale bar: 1 µm. (**L**) Computer 3D reconstruction of the SMLM double stained for βII-spectrin (red) and NMII heavy chain A (green) of the image shown in (**J**). Solid line follows the axon centerline, whereas the dashed line marks the cellular membrane. The red scale bar is 1.7 µm, which corresponds to the size of the βII-spectrin secondary structure (see panel O). A Z projection is shown. (**M**) Y projection related to (**J**). (**N**) Co-localization of βII-spectrin and NMII heavy chain A for the reconstruction shown in (**L**). (**Q**) Fluorescence intensity spatial frequency of βII-spectrin on the axonal axis, analyzed by Fourier transform. In addition to the 0.20 µm peak (marked with '#'), there is also a consistent peak at 1.7 µm (marked with '*'). (**P**) Models for the distribution of the NMII along axons. NMII filaments may crosslink adjacent actin rings (model a) or span individual rings (model b), in both cases with variable angles relative to the axonal axis.

resolution spinning disk microscopy to study differentiated SH-SY5Y (*Figure 4H*) and primary hippocampal neurons (*Figure 4I*). We show that in axons of both cell types, NMIIA can assemble into bipolar filaments of ~300 nm in length, consisting of eGFP puncta at the ends of the filament (N-terminus of head domains) with a single mApple punctum (C-terminus of tail domains) in the middle (*Figure 4H,I*). Similarly to the data obtained by STORM (*Figure 4D*), fluorescent NMII fusion proteins revealed the existence of multiple consecutive myosin filaments positioned along the axonal axis (*Figure 4H,I*).

Dual color SMLM for the simultaneous analysis of βII-spectrin and NMIIA, further suggested that NMII heavy chain A is mainly positioned within the axon core forming several filaments with different orientations in relation to the axon shaft (*Figure 4J*; *Video 2*). The analysis of the NMIIA labeling envelope, that is, the smooth curve outlining the extremes of NMIIA labeling, showed variable orientations with respect to the axonal longitudinal axis (*Figure 4K*). This supports that NMIIA filaments are not strictly aligned with the axonal axis. 3D reconstructions were performed to quantify spatial properties (*Figure 4L–N*). SMLM showed sites of colocalization of NMIIA heavy chain and βII-spectrin (*Figure 4N*), which suggests the existence of NMII filaments crosslinking adjacent rings (*Figure 4Q*, model a). Recently, immunogold labeling of pMLC followed by platinum-replica electron microscopy showed gold beads along filaments perpendicular to actin rings (*Vassilopoulos et al., 2019*). This led to the hypothesis that these filaments might correspond to myosins associated with the MPS, cross-linking neighboring rings, which goes along our observations. One should however note that in our analysis, some of the NMIIA staining was non-coincident with βII-spectrin, thus pointing towards a possible NMIIA filament distribution within actin rings (*Figure 4Q*, model b). As expected, βII-spectrin showed a prominent peak in the spatial frequency domain, corresponding to a 197 nm periodicity (*Figure 4O*, highlighted with #). Interestingly, a second pronounced peak at 1.71 µm was also observed (*Figure 4O*, highlighted with *), suggesting a possible secondary structure arrangement of βII-spectrin in axons, that should be the subject of further analysis.

## Structural organization and dynamics of actomyosin axonal rings

Our findings highlight possible distinct spatial positions of NMII filaments with respect to the MPS, which correlate with different biomechanical roles. NMII filaments crosslinking adjacent actin rings (*Figure 4P*, model a) are not expected to provide for radial contractility but provide for scaffolding. NMII filaments within individual actin rings (*Figure 4P*, model b), represent the conformation capable of generating the highest contractile force leading to actin filament sliding along the ring. Assuming that the MPS actin rings are composed of two parallel, intertwined

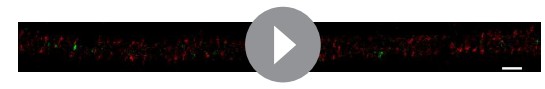

**Video 2.** 3D two-color SMLM of DIV8 hippocampal neurons immunostained against βII-spectrin (red) and NMIIA (green). Scale bar: 500 nm.
https://elifesciences.org/articles/55471#video2

actin filaments (*Vassilopoulos et al., 2019*), NMII activity could provide for contraction by putting the two actin filaments in registry with each other. One should however note that from the images presented (*Vassilopoulos et al., 2019*), a non-uniform distribution of NMII along the actin structures is observed which, to our understanding, cannot generate significant contractile force but rather hold a crosslinking function.

It is possible that in a single ring, NMII filaments with scaffolding and contractile roles might co-exist. Additionally, scaffolding and contractile roles may switch via change in activity of the NMII motors. Pharmacological NMII inhibition may lead to ring expansion given loss of both scaffolding and contractile roles; as a consequence, the inner axonal tension will lead to increased axon diameter. When axonal diameter is restored, contractile NMII filaments may perform ring contraction, favored by transient absence of scaffolding filaments. Eventually, scaffolding NMII may be restored and contraction may slow down to its original low rate. In physiological responses (such as axon diameter increase upon cargo trafficking) at an initial stage, scaffolding NMII may be reduced to a minimum, while contractile filaments cannot prevent ring expansion. At a second stage (when cargo moves), contractile NMII may lead to a fast return to the original diameter. Eventually, scaffolding filaments may be restored and the contraction rate will return to a very low value. One should however bear in mind that despite the fact that NMII activity modulates MPS diameter, and that pMLC colocalizes with MPS actin both in the AIS and in the axon shaft, the interplay between NMII and deep axonal actin filaments may also occur. In this scenario, the possibility that these alternative axonal actin arrangements may also serve as myosin anchors cannot be ruled out and should be certainly explored in the future. In summary, our data supports that NMII regulates circumferential axonal contraction and expansion. These findings have important implications on our understanding of neuronal biology, including fluctuations in axonal diameter observed during trafficking, action potential firing and axon degeneration.

# Materials and methods

**Key resources table**

| Reagent type (species) or resource | Designation | Source or reference | Identifiers | Additional information |
|---|---|---|---|---|
| Antibody | Rabbit polyclonal anti-NMIIA | Sigma-Aldrich | Cat# M8064, RRID:AB_260673 | WB_1:1000; IF_1:200 |
| Antibody | Rabbit polyclonal anti-NMIIB | Sigma-Aldrich | Cat# M7939, RRID:AB_260669 | WB_1:1000; IF_1:200 |
| Antibody | Rabbit polyclonal canti-α-adducin | Abcam | Cat# ab51130, RRID:AB_867519 | WB_1:1000 |
| Antibody | Rabbit monoclonal anti-vinculin | Abcam | Cat# ab129002, RRID:AB_11144129 | WB_1:1000 |
| Antibody | Mouse monoclonal anti-βactin | Sigma-Aldrich | Cat# A5441, RRID:AB_476744 | WB_1:5000 |
| Antibody | Mouse monoclonal anti-αtubulin | Sigma-Aldrich | Cat# T6199, RRID:AB_477583 | WB_1:1000 |
| Antibody | Peroxidase-AffiniPure donkey polyclonal anti-rabbit IgG (H+L) | Jackson Immuno Research Labs | Cat# 711-035-152, RRID:AB_10015282 | WB_1:5000 |
| Antibody | Peroxidase-AffiniPure donkey polyclonal anti-mouse IgG (H+L) | Jackson Immuno Research Labs | Cat# 715-035-151, RRID:AB_2340771 | WB_1:5000 |
| Antibody | Mouse monoclonal anti-βII-spectrin | BD Transduction | Cat# 612563, RRID:AB_399854 | IF_1:200 |
| Antibody | Rabbit polyclonal anti-MAP2 | Synaptic Systems | Cat# 188002, RRID:AB_2138183 | IF_1:20000 |
| Antibody | Rabbit polyclonal anti-NMIIC | Robert Adelstein NHLBI, Bethesda, USA | N/A | IF_1:40 |

*Continued on next page*

*Continued*

| Reagent type (species) or resource | Designation | Source or reference | Identifiers | Additional information |
|---|---|---|---|---|
| Antibody | Rabbit polyclonal anti-pMLC2 Thr18/Ser19 | Cell Signaling | Cat# 3674, RRID:AB_2147464 | IF_1:50 |
| Antibody | Rabbit polyclonal anti-MYPT1 | Cell Signaling | Cat# 2634, RRID:AB_915965 | IF_1:50 |
| Antibody | Rabbit monoclonal anti-MLC2 | Cell Signaling | Cat# 1678505S | IF_1:40 |
| Antibody | Goat polyclonal anti-mouse STAR 635P | Abberior GmbH | Cat# 2-0002-007-5 | IF_1:200 |
| Antibody | Goat polyclonal anti-rabbit STAR 635P | Abberior GmbH | Cat# 2-0012-007-2 | IF_1:200 |
| Antibody | Goat polyclonal anti-mouse STAR 580 | Abberior GmbH | Cat# 2-0002-005-1, RRID:AB_2620153 | IF_1:200 |
| Antibody | Goat polyclonal anti-mouse STAR 580 | Abberior GmbH | Cat# 2-0012-005-8 | IF_1:200 |
| Antibody | Goat polyclonal anti- mouse Alexa Fluor 532 | Thermo Fischer Scientific | Cat# A-11002, RRID:AB_2534070 | IF_1:200 |
| Antibody | Donkey polyclonal anti- rabbit Alexa Fluor 647 | Jackson ImmunoResearch | Cat# 711-605-152 | IF_1:1000 STORM_ 1:200 |
| Chemical compound, drug | Blebbistatin | Sigma-Aldrich | Cat# B0560 | 3 µM |
| Chemical compound, drug | ML-7 | Sigma-Aldrich | Cat# I2764 | 4 µM |
| Chemical compound, drug | Myovin1 | Calbiochem | Cat# 475984 | 4 µM |
| Chemical compound, drug | Calyculin A | Sigma-Aldrich | Cat# C5552 | 5 nM |
| Chemical compound, drug | Dimethyl sulfoxide (DMSO) | VWR International | Cat# A3672. 0050 | N/A |
| Chemical compound, drug | Phalloidin 635P | Abberior GmbH | Cat# 2-0205-002-5 | 0.33 µM |
| Cell line | CAD (mouse) | ECACC through Sigma-Aldrich | Cat# 08100805, RRID:CVCL_0199 | N/A |
| Cell line | PC-12 (rat) | ATCC | Cat# CTL-1721, RRID:CVCL_F659 | N/A |
| Cell line | SH-SY5Y (human) | ATCC | Cat# CRL-2266, RRID:CVCL_0019 | N/A |
| Recombinant DNA reagent | pEGFP-C1 | Addgene | N/A | N/A |
| Recombinant DNA reagent | pLKO.1 | Addgene | N/A | N/A |
| Recombinant DNA reagent | shRNA NMIIA | *Rai et al., 2017* | N/A | 5'GCGATACTACTCAGGGCTTAT3' |
| Recombinant DNA reagent | shRNA NMIIB | This paper | N/A | 5'GCCAACATTGAAACATACCT3' |

*Continued on next page*

*Continued*

| Reagent type (species) or resource | Designation | Source or reference | Identifiers | Additional information |
|---|---|---|---|---|
| Recombinant DNA reagent | shRNA NMIIC | This paper | N/A | 5'CCGGGCTCATTTATACCTACT3' |
| Recombinant DNA reagent | shRNA RLC | *Wang et al., 2008* | N/A | 5'GCACGGAGCGAAAGACAAA3' |
| Recombinant DNA reagent | shRNA MYPT1 | This paper | N/A | 5'GAGCCTTGATCAGAGTTATAAC3' |
| Recombinant DNA reagent | shRNA α-adducin | Sigma-Aldrich | Cat# TRCN0000108809 | 5'GCAGAAGAAGAGGGTGTCTAT3' |
| Recombinant DNA reagent | Human mutated ShRNA-resistant NMIIA | This paper | N/A | Vectorbuilder Page 20_line 374–376 |
| Recombinant DNA reagent | Human mutated ShRNA-resistant NMIIB | This paper | N/A | Vectorbuilder Page 20_line 374–376 |
| Recombinant DNA reagent | Human ShRNA-resistant RLC | Addgene | Cat #35680 | N/A |
| Recombinant DNA reagent | Human mutated ShRNA-resistant MYPT1 | This paper | N/A | Vectorbuilder Page 20_line 374–376 |
| Recombinant DNA reagent | CMV-eGFP-NMIIA | Addgene | Cat #11347 | N/A |
| Recombinant DNA reagent | CMV-NMIIA-mApple | John Hammer, NHLBI, Bethesda, USA | N/A | N/A |
| Commercial assay or kit | NZY Total RNA Isolation Kit | NZY Tech | Cat# MB13402 | N/A |
| Commercial assay or kit | SuperScript First-Strand Synthesis System for RT-PCR | Thermo Fisher Scientific | Cat# 11904018 | N/A |
| Sequence-based reagent | NMIIC sense primer | This paper | N/A | 5'CCTGGCTGAGTTCTCCTCAC3' |
| Sequence-based reagent | NMIIC antisense primer | This paper | N/A | 5'TGCTTCTGCTCCATCATCTG3' |
| Sequence-based reagent | RLC sense primer | This paper | N/A | 5'CCTTTGCCTGCTTTGATGAG3' |
| Sequence-based reagent | RLC antisense primer | This paper | N/A | 5'GTGACTGGGATGGGGTGTAG3' |
| Sequence-based reagent | MYPT1 sense primer | This paper | N/A | 5'AAGGGAACGAAGAGCTCTAGAAA3' |
| Sequence-based reagent | MYPT1 antisense primer | This paper | N/A | 5'TGACAGTCTCCAGGGGTTCT3' |
| Sequence-based reagent | β-actin sense primer | This paper | N/A | 5'ACCACACCTTCTACAATGAG3' |
| Sequence-based reagent | β-actin antisense primer | This paper | N/A | 5'TAGCACAGCCTGGATAGC3' |

*Continued on next page*

*Continued*

| Reagent type (species) or resource | Designation | Source or reference | Identifiers | Additional information |
|---|---|---|---|---|
| Sequence-based reagent | GADPH sense primer | This paper | N/A | 5'AGGCACCAAGATACTTACAAAAAC3' |
| Sequence-based reagent | GADPH antisense primer | This paper | N/A | 5'TGTATTGTAACCAGTCATCAGCA3' |
| Software, algorithm | MATLAB R2018a | MATLAB | RRID:SCR_001622 | N/A |
| Software, algorithm | Fiji | NIH | SRRID:SCR_002285 | N/A |
| Software, algorithm | Leica LAS X software | Leica | RRID:SCR_013673 | N/A |
| Software, algorithm | µSpikeHunter software | *Heiney et al., 2019* | N/A | N/A |
| Software, algorithm | GraphPad Prism | GraphPad | RRID:SCR_002798 | N/A |
| Software, algorithm | Huygens Software | Scientific Volume Imaging | RRID:SCR_014237 | N/A |

## Hippocampal neuron cultures

Mice and rat hippocampal neuron cultures were performed as described previously (*Kaech and Banker, 2006*). Briefly, the hippocampus of each individual E18 embryo was digested 15 min in 0.06% trypsin from porcine pancreas solution (Sigma-Aldrich, cat# T4799) and triturated. Either 12,500 cells/coverslip (for STED imaging) or 50,000 cells/coverslip (for SMLM imaging) were plated onto 50 µg/mL poly-L-lysine hydrobromide (Sigma-Aldrich, cat# P2636-100MG) pre-coated 1.5H glass 13 mm rounded coverslips (Marienfeld, for STED imaging) or 22 × 22 mm coverslips (Corning, for SMLM imaging) in 24- or 6-well plates (Nunc). Neurons were cultured in Neurobasal medium (Thermo Fisher Scientific, cat# 21103–049) supplemented with 2% B-27 (Thermo Fisher Scientific, cat# 0080085SA), 1% penicillin/streptomycin (Thermo Fisher Scientific, cat# 15140–122) and 2 mM L-glutamine (Thermo Fisher Scientific, cat# 25030024). In all conditions, cells were fixed with 4% (w/v) paraformaldehyde (PFA) in phosphate-buffered saline (PBS) at pH7.4 for 20 min at room temperature.

## Cell lines

CAD cells (mouse catecholaminergic neuronal cell line¸ ECACC through Sigma-Aldrich cat# 08100805; authenticated by DNA barcoding; mycoplasma contamination testing status: negative), PC-12 cells (rat adrenal gland pheochromocytoma, ATCC cat# CTL-1721; mycoplasma contamination testing status: negative) and SH-SY5Y cells (human neuroblastoma, ATCC cat# CRL-2266; mycoplasma contamination testing status: negative) were used in specific experiments as detailed below. Both CAD and PC-12 cells were maintained in Dulbecco's modified Eagle's medium (DMEM, Sigma-Aldrich, cat# D6429-500ML) supplemented with 10% fetal bovine serum (FBS, Sigma-Aldrich, cat#F9665-500ML), and 1% penicillin/streptomycin, while SH-SY5Y cells were cultured in DMEM:F12 (Sigma-Aldrich, cat# D8437−6 × 500 ML) 1:1, with 10% FBS, and 1% penicillin/streptomycin. SH-SY5Y were differentiated as detailed (*Encinas et al., 2002*). Briefly, 6500 cells were plated on 10 µg/mL poly-D-lysine hydrobromide (Sigma-Aldrich, cat# P0899) and 5 µg/mL laminin (Sigma-Aldrich, cat# L2020) pre-coated 1.5H glass 13 mm rounded coverslips (Marienfeld, for STED imaging) in a 24-well plate (Nunc). In the following day, media was supplemented with 10 µM retinoic acid (Sigma-Aldrich, cat# R2625) and renewed every other day for 5 days. At day 6, the media was changed for DMEM:F12 with 2% B-27, 1% penicillin/streptomycin and brain derived neurotrophic (BDNF, Sigma-Aldrich, cat# B3795, 50 ng/ml); at day 9 differentiated cells were used for imaging.

## Modulation of MNII activity using pharmacological agents

To modulate myosin activity, blebbistatin (Sigma-Aldrich, cat# B0560, 3 µM), ML-7 (Sigma-Aldrich, cat# I2764, 4 µM), or myovin1 (Calbiochem, cat# 475984, 4 µM), were added to hippocampal neurons at DIV3 and at DIV8 1 hr prior to fixation. Calyculin A (Sigma-Aldrich, cat# C5552, 5 nM) was added to hippocampal neurons at DIV8, 25 min prior to fixation. For all drugs, dimethyl sulfoxide (DMSO, VWR International, cat# A3672.0050) was used as a vehicle. In control conditions, vehicle alone was used at the same concentration as in the respective experimental condition. For the analysis of the effect of blebbistatin in in vitro thinning, the drug or its vehicle were additionally added at DIV16 and at DIV22 1 hr prior to fixation. To evaluate the reversibility of the effect of blebbistatin, at DIV8 the drug was either replaced by vehicle, or fresh drug was added to the culture. To confirm the inhibitory effect of ML-7 on RLC phosphorylation, hippocampal neurons were incubated with either vehicle or ML-7 (4 µM) at DIV8 1 hr prior fixation and immunofluorescence against pMLC was conducted as detailed below. To evaluate reversibility of the effect of calyculin A on axonal diameter, hippocampal neurons were treated with blebbistatin (3 µM) and calyculin A (5 nM) at DIV8; 25 min later the cells were fixed. To evaluate a possible cumulative effect of adducin and NMII in axonal diameter, sh-mediated knockdown of α-adducin was performed at DIV3 (detailed below) and either blebbistatin or vehicle were added at DIV4 and DIV8 1 hr before fixation. shRNA-mediated downregulation shRNA constructs against NMIIA, NMIIB, NMIIC, RLC, MYPT1 and α-adducin were used. Sequences were the following: for NMIIA 5'-GCGATACTACTCAGGGCTTAT-3' (*Rai et al., 2017*); for NMIIB 5'-GCCAACATTGAAACATACCT-3' (VectorBuilder); for NMIIC 5'-CCGGGCTCATTTATACCTACT-3' (VectorBuilder); for RLC 5'-GCACGGAGCGAAAGACAAA-3' (*Wang et al., 2008*); for MYPT1 5'-GAGCCTTGATCAGAGTTATAAC-3' (Vector Builder) and for α-adducin 5'-GCAGAAGAAGAGGGTGTCTAT-3' (TRCN0000108809, Sigma). In each case, rescue experiments were performed using shRNA-resistant constructs containing 2–3 mismatches in relation to the respective shRNA. For NMIIA, NMIIB and MYPT1 these were designed and ordered from Vector Builder; for RLC, the shRNA-resistant construct was obtained from Addgene (#35680). In control conditions, the pLKO.1 empty vector (Addgene) or a shRNA scramble plasmid (VectorBuilder) were used. At DIV3 hippocampal neurons were co-transfected with specific shRNAs and pEGFP-C1 (0.5 µg:0.25 µg/wells) or in the case of rescue experiments, additionally with the respective shRNA-resistant construct (0.5 µg/well) using Lipofectamine 3000 (Thermo Fisher Scientific, cat# L3000001) according to the manufacturer's instructions. Cells were fixed at DIV8. In all conditions, transfected (EGFP-positive) and non-transfected (EGFP-negative) cells in the same well were used for analysis. To validate shRNAs, CAD cells, PC-12 cells and hippocampal neurons were used. CAD and PC-12 cells were seeded at a density of 250000 cells/well while hippocampal neurons at a density of 12500 cells/well in 24-well plates and co-transfected with specific shRNAs and pEGFP-C1 (1.5 µg:0.5 µg) using Lipofectamine 2000 (Thermo Fisher Scientific, cat# 116678030) for cell lines, and Lipofectamine 3000 for hippocampal neurons. To evaluate shRNA efficiency, either Western blot analysis of CAD or PC-12 cells (for adducin, NMIIA and NMIIB) or RT-PCR (for NMIIC and MYPT-1) were performed. Additionally, immunofluorescence of primary hippocampal neurons was conducted for NMIIA, NMIIB, NMIIC, RLC and MYPT1, as detailed below. For immunoblotting, cell extracts were prepared in lysis buffer (0.3% Triton X-100 (Sigma-Aldrich, cat# T9284-100ML), protease inhibitors (cOmplete, Mini, Roche, Merck, cat# 04693124001) and 2 mM orthovanadate (Sigma-Aldrich, cat# S6508-10G) in PBS), run in 10% SDS-PAGE gels and transferred to Amersham Protran Premium 0.45 µm nitrocellulose membranes (GE Healthcare Life Sciences, VWR International, cat# GEHE10600003). The following primary antibodies were used: rabbit anti-NMIIA (Sigma-Aldrich, cat# M8064, 1:1000), rabbit anti-NMIIB (Sigma-Aldrich, cat# M7939, 1:1000), rabbit anti-α-adducin (Abcam, cat# ab51130, 1:1000); rabbit anti-vinculin (Abcam cat# ab129002, 1:1000), mouse anti-β-actin (Sigma-Aldrich, cat# A5441, 1:5000), and mouse anti-α-tubulin (Sigma-Aldrich, cat# T6199, 1:1000). The secondary antibodies donkey anti-rabbit IgG conjugated with horseradish peroxidase (HRP) (Jackson Immuno Research, cat# 711-035-152, 1:5000) and donkey anti-mouse IgG conjugated with HRP (Jackson Immuno Research, cat# 715-035-151, 1:5000) were employed. Quantifications were performed using Quantity One 1-D Analysis Software version 4.6 (Bio-Rad). In the case of RT-PCR, RNA from CAD and PC12 cells was extracted using NZY Total RNA Isolation Kit (NZY Tech, cat# MB13402). RNA concentration and purity were determined by NanoDrop spectrophotometry, and integrity was assessed using BioRad's Experion RNA chip. cDNA synthesis was performed with SuperScript First-Strand Synthesis System for RT-PCR

(Thermo Fisher Scientific, cat# 11904018). RT-PCR was done using the following specific primers: for NMIIC (forward 5'-CCTGGCTGAGTTCTCCTCAC-3' and reverse 5'-TGCTTCTGCTCCATCATCTG-3', for amplification of a 207 bp fragment); for MYPT1 (forward 5'-AAGGGAACGAAGAGCTCTAGAAA-3' and reverse 5'-TGACAGTCTCCAGGGGTTCT-3', for amplification of a 242 bp fragment; β-actin (forward 5'-ACCACACCTTCTACAATGAG-3' and reverse 5'-TAGCACAGCCTGGATAGC-3', for amplification of a 161 bp fragment) and GADPH (forward 5'-AGGCACCAAGATACTTACAAAAAC-3' and reverse 5'- TGTATTGTAACCAGTCATCAGCA-3', for amplification of a 193 bp fragment).

## Immunolabeling

Primary hippocampal neurons were fixed with 4% PFA, in PBS at pH 7.4 for 20 min at room temperature. Fixed cells were permeabilized with 0.1% (v/v) triton X-100 (in PBS) for 5 min and autofluorescence was quenched with 0.2M ammonium chloride (Merck, cat# 1.01145.0500). Non-specific labeling was blocked by incubation with blocking buffer (5% FBS in PBS) for 1 hr. Primary antibodies diluted in blocking buffer were incubated overnight at 4°C: mouse anti-βII-spectrin (BD Transduction, cat# 612563, 1:200), rabbit anti-MAP2 (Synaptic Systems, cat# 188002, 1:20000), rabbit anti-NMIIA (Sigma-Aldrich, cat# M8064, 1:200), rabbit anti-NMIIB (Sigma-Aldrich, cat# M7939, 1:200), rabbit anti-NMIIC (a kind gift from Dr Robert Adelstein, 1:40), rabbit anti-pMLC2 Thr18/Ser19 (Cell Signaling, cat# 3674, 1:50), rabbit anti-MLC2 (Cell Signaling, cat# 1678505S, 1:40), and rabbit anti-MYPT1 (Cell Signaling, cat# 2634, 1:50). After three 5 min washes in PBS, incubation with secondary antibody was performed for 1 hr at room temperature. For STED microscopy, the following secondary antibodies were used: goat anti-mouse STAR 635P (Abberior GmbH, cat# 2-0002-007-5, 1:200); goat anti-rabbit STAR 635P (Abberior GmbH, cat# 2-0012-007-2, 1:200); goat anti-mouse STAR 580 (Abberior GmbH, cat# 2-0002-005-1, 1:200) and goat anti-rabbit STAR 580 (Abberior GmbH, cat# 2-0012-005-8, 1:200). For actin staining, 0.3 µM phalloidin 635P (Abberior GmbH, cat# 2-0205-002-5, in PBS), was used. Phalloidin was incubated in cells for 1 hr at 37°C following the secondary antibody incubation. For SMLM dSTORM the following secondaries were used: goat-anti mouse Alexa Fluor 532 (Thermo Fischer Scientific, cat# A-11002, 1:200) and donkey-anti rabbit Alexa Fluor 647 (Jackson ImmunoResearch, cat# 711-605-152, 1:200). After three 5 min PBS washes, coverslips were mounted in 80% glycerol for STED microscopy; for SMLM imaging, coverslips were mounted in GLOX/MEA buffer (detailed below) in depression slides and sealed with twinsil (Picodent, cat# 13001000) and for immunofluorescence, the coverslips were mounted in fluoroshield with DAPI (Vector Laboratories, cat# H-1200) and sealed with nail varnish. In the case of shRNA validations, images were acquired using a Zeiss Axio ImagerZ1 widefield microscope (Carl Zeiss) equipped with oil-immersion 63x/1.4 (Plan-Apochromat) and with Differential Interference Contrast (DIC) and a TCS Leica SP8 confocal microscope (Leica Microsystems). To validate the effectiveness of shRNA-mediated downregulation of NMIIA, NMIIB, NMIIC, RLC and MYPT1, the cell body from transfected and non-transfected cells, was delineated with the segmented line tool from Fiji, and the mean fluorescence intensity was measured. To confirm that ML-7 inhibits the phosphorylation of NMII, the AIS was delineated with the segmented line from Fiji and the mean fluorescence intensity was compared in treated and non-treated cells.

## STED imaging

STED imaging was performed on an inverted Leica TCS SP8 STED 3X (Leica Microsystems), using DIV8 hippocampal neurons, unless otherwise indicated. Hippocampal neurons were imaged, at a fixed distance of 80–100 µm from the cell body, with a HC PLAPO CS2 100x NA 1.4 STED WHITE oil immersion objective (Leica Microsystems) using confocal and STED modes. The 2D vortex STED images with lateral resolution enhancement were recorded with 20 nm pixel size in xy and dwell times of typically 600 ns. First, the STED far-red channel (Abberior STAR 635P) was recorded with 633 nm excitation using the pulsed white light laser with 80 MHz repetition rate and STED depletion was performed with a synchronized pulsed 775 nm depletion laser. The detection bandpass was set to 650 to 750 nm and the pinhole was set to 0.93AU. The following acquisition settings were applied: 16 x line averaging and detector gating on a Hybrid Detector (HyD, Leica Microsystems) of 0.3 ns to 6 ns. The second STED channel (Abberior STAR580) was recorded in line sequential mode with 561 nm excitation and 775 nm depletion using a detection window from 580 to 620 nm. All other settings remained constant. We alternatively used an Abberior Instruments 'Expert Line'

gated-STED coupled to a Nikon Ti microscope with an oil-immersion 60x 1.4 NA Plan-Apo objective (Nikon, Lambda Series) and a pinhole size set at 0.8 Airy units. The system features 40 MHz modulated excitation (405, 488, 560 and 640 nm) and depletion (775 nm) lasers. The microscope's detectors are avalanche photodiode detectors (APDs) which were used to gate the detection between ~700 ps and 8ns. To analyze ring periodicity, the maximum intensity of peaks was determined and the interpeak distance was measured. To determine axon diameter, the distance between the outer points (brighter, in the focus plane) that formed the MPS was determined. Only axons unequivocally focused in the maximum wide plan were considered. Under distinct control conditions axon diameter varied, which is probably inherent to the different cultures used throughout the study. Given the use of specific controls in each experimental setting, this variation did not interfere with the interpretation of results. The tilting of actin rings was determined by measuring the angle of each actin ring regarding the axonal axis using the angle tool from Fiji. The tilting angle $\alpha$ was measured relative to the longitudinal axon axis. Angles larger than 90° were mirrored to the first quadrant to yield an effective angle $\alpha_{eff}=90-|90-\alpha|$.

## SMLM imaging with dSTORM/GSDIM

For super-resolution SMLM-imaging with the Leica SR GSD using the dSTORM/GSDIM protocol, 18 mm coverslips (50000 cells/slide) were stored in PBS after fixation and immunolabelling at 4°C. The coverslips were mounted onto a single depression slide (76 mm $\times$26 mm) and the cavity filled with 90–100 µl GLOX-MEA buffer (0.5 mg/ml glucose oxidase (Sigma-Aldrich, cat# G7141, 40 µg/ml catalase (Sigma-Aldrich, cat# 02071) 10% w/v glucose (Sigma-Aldrich, cat# 49163), 50 mM Tris-HCl pH 8.0, 10 mM NaCl and 10 mM β-mercaptoethylamine (Sigma-Aldrich, cat# M9768-5G)). The buffer was freshly prepared before imaging. Imaging was performed with a Leica SR GSD system using a HC PL APO 160×/NA 1.43 oil objective. The images were recorded with an Andor iXon 897 EMCCD camera at 40 Hz using a central 180 pixel x 180 pixel subregion. For excitation, a 532 nm laser (500 mW maximum power output) and a 642 nm laser (500 mW maximum power output) were used and attenuated using an AOTF when appropriate. The two fluorophores were recorded sequentially and image acquisition, single molecule analysis and image reconstruction was performed with Leica LAS X 1.9.0.13747.

## Spinning disk imaging

Differentiated SH-SY5Y cells (6500 cells/well) and DIV6-hippocampal neurons (50000 cells/well) were co-transfected with CMV-eGFP-NMIIA (Addgene, cat# 11347) and NMIIA-mApple (a kind gift from Dr John Hammer) using 1 µg:1 µg of each construct/well and Lipofectamine 3000 following the manufacturer's instructions. Two days later, in the case of SH-SY5Y cells, and four days later (at DIV10) in the case of primary hippocampal neurons, the cells were fixed. Transfected cells were then imaged using an Olympus SpinSR10 spinning disk confocal super-resolution microscope (Olympus, Tokyo, Japan) equipped with an PlanAPON 60 ×/1.42 NA oil objective (Olympus), a CSU-W1 SoRa-Unit (Yokogawa, Tokyo, Japan) with 3.2x magnification and ORCA-Flash 4.0 V3 Digital CMOS Camera (Hamamatsu, Hamamatsu City, Japan).

## Preparation of microelectrode–microfluidic devices and electrophysiology recordings

Custom designed µEF devices were prepared following *Lopes et al. (2018)*. Briefly, coated MEA chips (MultiChannel Systems MCS GmbH, Germany), with 252 recording electrodes of 30 µm in diameter and a center-to-center inter-electrode spacing of 100 µm, were combined with polydimethylsiloxane (PDMS) microfluidic chambers with an appropriate microgroove spacing for compartmentalization and monitoring of axonal activity. MEAs were coated with 0.01 mg/ml of poly-D-lysine (PDL, Corning) overnight at 37°C, and then washed with sterile water. Microfluidic devices were sterilized with 70% ethanol and were gently attached to PDL-coated MEAs, creating a µEF chamber composed of two separate compartments connected by 700 µm length $\times$9.6 µm height $\times$10 µm width microchannels. The medium reservoirs were loaded with 150 µl of 5 µg/ml laminin isolated from mouse Engelbreth-Holm-Swarm sarcoma (Sigma-Aldrich Co.) and incubated overnight at 37°C. The unbound laminin-1 was removed, and chambers were refilled with Neurobasal medium and left to equilibrate for at least 2 hr at 37°C before cell seeding. Hippocampal neurons at DIVs 11, 12 and

14 were used in the electrophysiology experiments, where either blebbistatin (3 µM), or vehicle were added. Recordings were performed using a MEA2100 recording system (MultiChannel Systems MCS GmbH, Germany). The µEF devices prepared with these MEAs had 16 microchannels with 7 electrodes positioned along each microchannel, as well as 126 electrodes dedicated to the somal compartment. For each time point, recordings were obtained at a sampling rate of 20 kHz for the characterization of the overall network activity. Then, high-temporal resolution recordings were obtained at a sampling rate of 50 kHz, for a duration of 60 or 120 s, for the calculation of the conduction velocity. Throughout the experiments, the temperature was maintained at 37°C and all recorded activity was spontaneous activity. Data analysis was carried out in MATLAB R2018a (The MathWorks Inc) using custom scripts (available in GitHub at: https://github.com/paulodecastroaguiar/Calculate_APs_velocities_in_MEAs; copy archived at https://github.com/elifesciences-publications/Calculate_APs_velocities_in_MEAs; *Aguiar, 2020*) and the µSpikeHunter tool (*Heiney et al., 2019*). Raw signals were band-pass filtered (200–3000 Hz) and spikes were detected by a threshold set to $6 \times$ STD of the electrode noise. Electrodes with a mean firing rate (MFR) of at least 0.1 Hz were considered active. For the propagation velocity calculations, the extracted spike times were further corrected based on the voltage waveforms. To be considered a propagating event the following requirements had to be fulfilled: detection over the entire microchannel (7 electrodes); time delay between electrode pairs lower than or equal to 1 ms (minimum propagation velocity of 0.1 m/s); isolated spike in a 3 ms time window (as to ensure spike identity in all electrodes). Propagation velocity was then calculated by dividing the first-to-last electrode distance (600 µm span) by the delay between spike times. This stringent detection method eliminated any ambiguity during bursts and excluded sequences with missing spike times on at least one electrode, which drastically reduced the size but strengthened the quality of the action potentials dataset.

## Statistical analysis

All measurements were performed with the researcher blinded to the experimental condition. Data are shown as mean ± s.e.m, which the exception of propagation velocity values which are shown as median ± s.d. Statistical significance was determined by Student's t-test using Prism (GraphPad Software), with exception of actin ring measurements in hippocampal neuron cultures, where one-way ANOVA was used (GraphPad Software). Sample sizes are indicated in Figure legends and significance was defined as $p^*<0.05$, $p^{**}<0.01$, $p^{***}<0.001$, $p^{****}<0.0001$, ns – not significant.

## Acknowledgements

We thank Dr Sandra Sousa (Molecular Microbiology IBMC/i3S), Dr Robert S Adelstein and Dr John A Hammer (National Heart, Lung, and Blood Institute, Bethesda, MD) for reagents, the Advanced Light Microscopy Facility at EMBL, especially Dr A Halavatyi for R-scripts, the Advanced Light Microscopy Facility at IBMC/i3S (PPBI-POCI-01–0145-FEDER-022122), Dr Hélder Maiato for support in microscopy (Chromosome Instability and Dynamics IBMC/i3S) and Leica Microsystems for support. This work was financed by FEDER - Fundo Europeu de Desenvolvimento Regional funds through the NORTE 2020 - Norte Portugal Regional Operational Programme, Portugal 2020, and by Portuguese funds through FCT - Fundação para a Ciência e a Tecnologia/Ministério da Ciência, Tecnologia e Ensino Superior in the framework of the project NORTE-01-0145-FEDER-028623 (PTDC/MED-NEU/28623/2017). The project was also supported by the Infrastructure for NMR, EM and X-rays for Translational Research- iNEXT. Work done with AJP was funded by a grant agreement from the European Research Council (681443) under the European Union's Horizon 2020 research and innovation program. ARC, SCS and JCM are funded by FCT (SFRH/BPD/114912/2016, SFRH/BD/136760/2018 and PD/BD/135491/2018, respectively). SCS was additionally funded by the Christian Boulin Fellowships (EMBL, Heidelberg).

## Additional information

### Funding

| Funder | Grant reference number | Author |
|---|---|---|
| Fundação para a Ciência e a Tecnologia | NORTE-01-0145-FEDER-028623 | Monica M Sousa |
| Fundação para a Ciência e a Tecnologia | PTDC/MED-NEU/28623/2017 | Monica M Sousa |
| European Research Council | 681443 | António J Pereira |
| FCT | SFRH/BPD/114912/2016 | Ana Rita Costa |
| FCT | SFRH/BD/136760/2018 | Sara C Sousa |
| FCT | PD/BD/135491/2018 | José C Mateus |
| EMBL Heidelberg | Christian Boulin Fellowships | Sara C Sousa |

The funders had no role in study design, data collection and interpretation, or the decision to submit the work for publication.

### Author contributions

Ana Rita Costa, Conceptualization, Data curation, Formal analysis, Validation, Investigation, Visualization, Writing - original draft, Writing - review and editing; Sara C Sousa, Rita Pinto-Costa, Ana Catarina Costa, David Rosa, Diana Machado, Luis Pajuelo, Xuewei Wang, Investigation; José C Mateus, Data curation, Formal analysis, Validation, Investigation, Performed electrophysiology; Cátia DF Lopes, Investigation, Performed electrophysiology; Feng-quan Zhou, Resources, Supervision, Writing - review and editing; António J Pereira, Software, Visualization, Writing - review and editing; Paula Sampaio, Formal analysis, Writing - review and editing; Boris Y Rubinstein, Inês Mendes Pinto, Formal analysis, Writing - original draft, Writing - review and editing, Analysed the biomechanical properties of different actomyosin conformations; Marko Lampe, Formal analysis, Supervision, Methodology, Writing - original draft, Writing - review and editing; Paulo Aguiar, Conceptualization, Software, Formal analysis, Supervision, Writing - original draft, Writing - review and editing; Monica M Sousa, Conceptualization, Formal analysis, Supervision, Funding acquisition, Methodology, Writing - original draft, Writing - review and editing

### Author ORCIDs

José C Mateus (iD) http://orcid.org/0000-0001-8058-5093
Xuewei Wang (iD) http://orcid.org/0000-0002-1375-7358
Marko Lampe (iD) http://orcid.org/0000-0002-4510-9048
Paulo Aguiar (iD) http://orcid.org/0000-0003-4164-5713
Monica M Sousa (iD) https://orcid.org/0000-0002-4524-2260

### Ethics

Animal experimentation: Experiments were carried out in accordance with the European Union Directive 2010/63/EU and national Decreto-lei n°113-2013. The protocols described were approved by the IBMC Ethical Committee and by the Portuguese Veterinarian Board.

### Decision letter and Author response

Decision letter https://doi.org/10.7554/eLife.55471.sa1
Author response https://doi.org/10.7554/eLife.55471.sa2

## Additional files

### Supplementary files

- Transparent reporting form

## Data availability

All data generated or analysed during this study are included in the manuscript and supporting files.

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
