## [Decision Letter]

**Acceptance summary:**

The mature axon shaft of a neuron is supported by a sub-membraneous actin-spectrin network, composed of actin rings regularly spaced by spectrin tetramers. This structure may provide mechanical support for axons, but the precise molecular composition of this sub-membraneous actin-spectrin network, and it’s similarities/differences to actin filament structures of other cell-types have remained elusive. This study reports that non-muscle myosin II (NMII) localizes to actin/spectrin rings of neuronal membrane periodic skeleton, and is important for regulating the expansion and contraction of axonal diameter. Moreover, by using super-resolution microscopy approaches, the authors provide evidence that NMII heavy chains are often present along the longitudinal axonal axis, being able to crosslink adjacent rings. Together, this study provides important new information on the molecular composition and functions of the membrane periodic skeleton of axons.

**Decision letter after peer review:**

Thank you for submitting your work entitled "The membrane periodic skeleton is an actomyosin network that regulates axonal diameter and conduction" for consideration by *eLife*. Your article has been reviewed by three peer reviewers, one of whom is a member of our Board of Reviewing Editors, and the evaluation has been overseen by a Senior Editor. The following individuals involved in review of your submission have agreed to reveal their identity: Pirta Hotulainen (Reviewer #2); Dylan Tyler Burnette (Reviewer #3).

Our decision has been reached after consultation between the reviewers. Based on these discussions and the individual reviews below, we regret to inform you that your work will not be considered further for publication in *eLife*, at least in its present form.

This manuscript reports that non-muscle myosin II (NMII) localizes to actin/spectrin rings of neuronal membrane periodic skeleton, and is important in regulating the expansion and contraction of axonal diameter. The findings presented in the manuscript are potentially of significant importance in the neurobiology field.

However, the reviewers felt that the shRNA experiments lacked the necessary controls. Moreover, the super-resolution imaging experiments aiming to uncover the localization and orientation of NMII bundles in axons were too preliminary at this stage. Thus, an extensive amount of additional work would be required to address these points. *eLife*

Reviewers, however, provide some suggestions for how to improve the manuscript. Thus, if you can address these issues by performing much better shRNA controls, as well as by significantly improving the imaging to determine the localization and orientation of NMII filaments, we would be glad to consider a new submission on this topic for publication in *eLife*. In this case, the new submission would be evaluated by the three original reviewers.

Reviewer #1:

This manuscript reports that non-muscle myosin II (NMII) localizes to actin/spectrin rings of neuronal membrane periodic skeleton, and is important for regulating the expansion and contraction of axonal diameter. The findings presented in the manuscript are potentially of significant importance in the neurobiology field, but in my opinion the second part of the manuscript focusing on the localization and orientation of NMII molecules in axons is too preliminary for publication in its present form. Thus, significant amount of additional imaging experiments aiming to better unravel the orientation of NMII molecules in axonal membrane periodic skeleton are required to strengthen the study.

Essential revisions:

1) While the periodic ring structure is clearly visible with BII-spectrin immunostaining (Figure 1), the periodicity is not obvious from pMLC and NMIIA/NMIIB antibody stainings presented in Figures 3 and Figure 4. Moreover, due to the bipolar nature of NMII filaments, pMLC should be visible as doublets with a spacing of 200-300 nm (see e.g. Beach et al., 2014; Fenix et al., 2016), but this is not obvious from the images presented in Figure 3. To more convincingly detect the localization of NMII in axons, the authors could perhaps consider using fluorescent NMII fusion proteins for these studies.

2) The NMII imaging experiments would be much more informative if the authors could simultaneously visualize the myosin motor domains and rods (see e.g. Fenix et al., 2016). Such experiments would provide more information about the actual orientation of NMII molecules along the axon, and help distinguishing between the possible orientations presented in Figure 4K.

3) In a schematic model (Figure 4K) the lengths of NMII filaments are highly variable. This does not match with the published literature, which to my knowledge suggest that the NMIIA and NMIIB bundles have regular lengths of approximately 300 nm (e.g. Billington et al., 2013). Thus, the authors should revise the model by taking into account the regular lengths of NMII filaments.

Reviewer #2:

Costa et al. manuscript is a short but carefully done report describing the role of acto-myosin contractility in the regulation of axonal diameter. Authors have used two molecules inhibiting myosin II contractility, one drug to activate myosin II contractility and one negative control drug against myoV contractility. Drug experiments were confirmed using shRNA approach and shRNAs were also tested against many units of myosin II complex, namely: myosin heavy chain, regulatory light chain and protein dephosphorylating (inactivating) myosin light chain (knock-down results more contractile myosin II). Authors also tested which of three myosin II isoforms, A, B or C, were responsible for axonal contractility (A and B positive, C not). All these attempts show the same result, myosin II contractility causes shrinking of axons. When the contractility is released by the way or another, axon diameter increases. To me, these results look very convincing.

1) Figure 1:

A. It is not clear what exactly the model on the right shows. If it visualizes contractility, actin filaments should slide closer to each other but here myosin II complex jumps forward.

L. and M. Myosin II manipulations did not affect periodicity of actin rings – but how about tilting of actin rings? I am not sure if tilting can be analyzed from STED images (how "straight" lines (actin rings) are) but I would be curious to see the result. I hope that it gives an insight of the directionality of myosin II contractility. If myosin II contracts inside rings, there should not be any tilting, but if myosin II contracts between rings, rings might tilt (change their angle).

L and M. are labeled as describing periodicity but to be exact, the analysis measures the average distance between peaks, not periodicity.

Figure legend mentions number of axons but it is not clear how many independent experiments these axons were taken from.

2) Figure 1—figure supplement 1: the title says: "Modulation of NMII activity regulates both the expansion and the contraction of axonal diameter", but the figure doesn't mention anything about the diameter of axons. Maybe good to change the title of the Figure.

3) Figure 2: Authors speculate that " One possibility is that inhibition of NMII activity might also affect the distribution of axonal sodium channels." This would be relatively easy to test by antibody staining of sodium channels and analysis of distribution. This experiment is optional.

4) Figure 4:

4B is dim and blurry, any possibilities to have a better image of phalloidin + myosin IIA?

4C: Just commenting that NMIIB staining looks exactly as it looks in our laboratory. For some reason, field expects that myosin II would show nice repeating periodicity but it does not. Staining looks a bit scattered as shown here, although there is also periodicity. I think we need to accept this and start to elucidate what does it mean.

4G XZ projection and 4J: where is the cell surface and inner "mid-point"? Give directions.

5) Figure 3:

I cannot see the concluded periodicity for pMLC from figures. To me it looks similar to myosin II heavy chains and my conclusion would be that pMLC and heavy chain are in complex and myosin II complex can be in different orientations. It would be interesting to have pMLC+ heavy chain together in super-resolution. Image might be complicated depending on the sites of labels but at same time, it could give an idea for the orientation of the complex. Would co-imaging of pMLC and myosin heavy chain be possible?

6) Figure 4K: Model figure: Here are many possibilities. I would ask authors to consider the length of myosin II complexes (bipolar (=300 nm)/ unipolar) and average diameter of an axon ring (here 400 nm, enlarged to 500 nm when contractility was abolished) to get reality to models. As much as I understand, myosin II complex is not changing its length, myosin II heavy chains are not sliding along each other as shown in the model figure.

If d) is the proposed model, I would expect that in xy, myosin II staining should follow the periodicity of actin rings but it does not. To me it seems that there is definitively something between rings. Is it sure that myosin II indeed attach actin filaments in actin rings? Are there other alternative actin filament structures in axons to bind?

Reviewer #3:

Costa et al., present evidence that myosin II contractility can regulate the diameter of neuronal axons. In addition, they provide striking super resolution imaging that support the proposed orientations of myosin II filaments in relation to the periodic actin rings along axons. This study is of potential interest. However, the authors have failed to provide controls for the vast majority of their experiments, which makes their implications difficult to interpret.

1) The authors use hsRNA to knockdown myosin II paralogs. Maybe I somehow missed it but there does not seem to be any evidence presented that the shRNA knockdown the myosin II paralogs in hippocampal neurons. Showing that they work in other cells (i.e., PC-12 or CAD) is not sufficient to make the claim that they are knockdown in primary cultured neurons. The authors should report how much is each myosin II paralog knocked down?

2) The authors make no attempt to show that phenotype reported for each shRNA knockdown is specifically because of the loss of the targeted paralog. The authors should reintroduce full length versions of each targeted paralog to test if it rescues the phenotype and relate the expression amount to endogenous levels in wild type axons. That is, does reintroduction lead to smaller axons?

3) By presented the aforementioned controls for the shRNA knockdowns, the authors will add sufficient support for the use of blebbistatin. However, there is also currently no controls presented to support their claim that ML-7 is inhibiting MLCK in hippocampal neurons. Is the phosphorylation of myosin light chain reduced in the neurons (either mono or di-phosphorylation)? In addition, it would be advisable to test if the ROCK inhibitor, Y-27632, also induces wider axons.

4) The authors claim that calyculin A is a protein phosphatase 1 (PP1) inhibitor. While this is the case, the omission that it is also a PP2A inhibitor implies a specificity that does not exist. As such, it is advisable to control for the use calyculin A. At the very least, it should be demonstrated that blebbistatin is able to mask the phenotypes induced by calyculin A.

5) What is the evidence that myosin V is being inhibited in the neurons by Myovin-1?

6) It does not seem to me that the singular use of blebbistatin is enough to make any claims as to the role of myosin II in regulating propagation velocity. This data seems preliminary at best.

---

## [Author Response]

Reviewer #1:This manuscript reports that non-muscle myosin II (NMII) localizes to actin/spectrin rings of neuronal membrane periodic skeleton, and is important for regulating the expansion and contraction of axonal diameter. The findings presented in the manuscript are potentially of significant importance in the neurobiology field, but in my opinion the second part of the manuscript focusing on the localization and orientation of NMII molecules in axons is too preliminary for publication in its present form. Thus, significant amount of additional imaging experiments aiming to better unravel the orientation of NMII molecules in axonal membrane periodic skeleton are required to strengthen the study.Essential revisions:1) While the periodic ring structure is clearly visible with BII-spectrin immunostaining (Figure 1), the periodicity is not obvious from pMLC and NMIIA/NMIIB antibody stainings presented in Figure 3 and Figure 4. Moreover, due to the bipolar nature of NMII filaments, pMLC should be visible as doublets with a spacing of 200-300 nm (see e.g. Beach et al., 2014; Fenix et al., 2016), but this is not obvious from the images presented in Figure 3. To more convincingly detect the localization of NMII in axons, the authors could perhaps consider using fluorescent NMII fusion proteins for these studies.

pMLC is expected to have a periodicity of ~190 nm if myosin is anchored to actin rings. Please note that this periodicity is still compatible with the existence of ~300 nm bipolar NMII filaments in axons (Figure 4O). A 190 nm periodic pMLC staining, coincident with phalloidin, was previously shown restricted to the AIS (Berger et al., 2018). This observation suggested that in the AIS, pMLC is bound to actin rings. Here we show that pMLC extends to the axon shaft (Figure 3D-F). Interestingly, in some axonal regions, pMLC showed a periodic distribution consistent with anchoring of myosin to different regions of consecutive actin rings (Figure 3D- highlighted by red ruler; and Figure 3F). The absence of a generalized striped pattern for pMLC (as is observed for axonal actin and βII-spectrin staining), supports that in each actin ring a limited number of pMLC molecules is anchored. Alternatively, technical limitations of the antibodies used may preclude the visualization of the entire pool of pMLC molecules present in the structure. Nonetheless, we now provide new imaging data showing that in some regions of the axon shaft, pMLC can be observed with the expected striped pattern, colocalized with actin (Figure3G). This supports that pMLC is bound to actin rings throughout the axon shaft. This issue is further discussed in the revised version of the manuscript (subsection “Phosphorylated NMII light chains are colocalized with actin within the MPS and organized as circular periodic structures persisting throughout the axon shaft”)

In the case of NMII heavy chains, please note that we imaged it using antibodies that recognize the middle portion of the NMII bipolar filament (Figure 4A-D). Unless NMII heavy chains would be positioned only within individual actin rings, a periodic pattern coincident with phalloidin is not expected. As suggested by the Reviewer, hippocampal neurons and the neuroblastoma cell line SH-SY5Y were transfected with fluorescent NMII fusion constructs that allow visualizing simultaneously the N- (eGFP tag) and C-terminal (mApple tag) of NMII heavy chain (Beach et al., 2014). NMII was then visualized using spinning disk super-resolution microscopy. In the revised version of the manuscript, we show that in axons, NMIIA can assemble into bipolar filaments of ~300 nm in length (Figure 4H and 4I). Similarly, to the data obtained by STORM (Figure 4D), fluorescent NMII fusion proteins revealed the existence of multiple consecutive myosin filaments, with different orientations in relation to the axonal axis. These results are presented and discussed in subsection “NMII heavy chains have multiple orientations in relation to individual actin rings and to the axonal axis, being able to crosslink adjacent rings”.

2) The NMII imaging experiments would be much more informative if the authors could simultaneously visualize the myosin motor domains and rods (see e.g. Fenix et al., 2016). Such experiments would provide more information about the actual orientation of NMII molecules along the axon, and help distinguishing between the possible orientations presented in Figure 4K.

As suggested by the reviewer and detailed above, we have imaged axonal myosin after transfection with fluorescent NMII fusion constructs (Figure 4 H and 4I). Our data shows the existence of bipolar NMII filaments of ~300 nm in length positioned along the axonal axis (supporting our previous STORM analysis of the NMII heavy chain- Figure 4D). Although the simultaneous observation of the two NMII fluorophores and either actin or βII-spectrin staining was technically not possible to implement, this new data demonstrates that multiple consecutive myosin filaments are positioned along the axonal axis.

3) In a schematic model (Figure 4K) the lengths of NMII filaments are highly variable. This does not match with the published literature, which to my knowledge suggest that the NMIIA and NMIIB bundles have regular lengths of approximately 300 nm (e.g. Billington et al., 2013). Thus, the authors should revise the model by taking into account the regular lengths of NMII filaments.

As requested by the reviewer, the models have been revised to accommodate the regular maximum lengths of approximately 300 nm of NMII filaments as well as an average axonal diameter of 450 nm.

Reviewer #2:Costa et al. manuscript is a short but carefully done report describing the role of acto-myosin contractility in the regulation of axonal diameter. Authors have used two molecules inhibiting myosin II contractility, one drug to activate myosin II contractility and one negative control drug against myoV contractility. Drug experiments were confirmed using shRNA approach and shRNAs were also tested against many units of myosin II complex, namely: myosin heavy chain, regulatory light chain and protein dephosphorylating (inactivating) myosin light chain (knock-down results more contractile myosin II). Authors also tested which of three myosin II isoforms, A, B or C, were responsible for axonal contractility (A and B positive, C not). All these attempts show the same result, myosin II contractility causes shrinking of axons. When the contractility is released by the way or another, axon diameter increases. To me, these results look very convincing.1) Figure 1:A. It is not clear what exactly the model on the right shows. If it visualizes contractility, actin filaments should slide closer to each other but here myosin II complex jumps forward.

As pointed out by the reviewer, to represent contractility, actin filaments have placed closer to each other.

L. and M. Myosin II manipulations did not affect periodicity of actin rings – but how about tilting of actin rings? I am not sure if tilting can be analyzed from STED images (how "straight" lines (actin rings) are) but I would be curious to see the result. I hope that it gives an insight of the directionality of myosin II contractility. If myosin II contracts inside rings, there should not be any tilting, but if myosin II contracts between rings, rings might tilt (change their angle).

As suggested by the reviewer, the possible tilting of actin rings has been measured for all drug treatments and Sh downregulations (Figure 1L and M of the revised version of the manuscript). A tilting angle D was measured relative to the longitudinal axon axis. Angles larger than 90º were mirrored to the first quadrant to yield an effective angle D_eff_ 90-|90-D_ In the absence of this correction, the average value would always artificially tend to 90 degrees. No variations were found, suggesting that myosin II probably does not contract between adjacent rings. However, it is also possible that adjacent actin rings may not tilt given the βII-spectrin scaffold. We also highlight that it is possible that having several active NMII filaments attached to the adjacent axon rings, the generated net force would be averaged out and thus close to zero. This has been added to subsection “Modulation of NMII activity regulates the expansion and contraction of axonal diameter”.

L and M. are labeled as describing periodicity but to be exact, the analysis measures the average distance between peaks, not periodicity.

As suggested, in the y axis of the graphs Figure 1L and M (now Figure 1N and O), “Periodicity” has been replaced by “Average distance between peaks”.

Figure legend mentions number of axons but it is not clear how many independent experiments these axons were taken from.

In every experiment displayed, at least 3 independent experiments have been performed. This is now stated in the figure legend.

2) Figure 1—figure supplement 1: the title says: "Modulation of NMII activity regulates both the expansion and the contraction of axonal diameter", but the figure doesn't mention anything about the diameter of axons. Maybe good to change the title of the Figure.

The title of Figure 1—figure supplement 1 has been changed to: “Analysis of ML-7 activity and ShRNA-mediated downregulation.”

3) Figure 2: Authors speculate that " One possibility is that inhibition of NMII activity might also affect the distribution of axonal sodium channels." This would be relatively easy to test by antibody staining of sodium channels and analysis of distribution. This experiment is optional.

Given the optional character of this experiment, this has not been included. However, this issue is discussed in subsection “Propagation velocity is altered by manipulation of NMII activity”.

4) Figure 4:4B is dim and blurry, any possibilities to have a better image of phalloidin + myosin IIA?

Figure 4B has been changed and an image of better quality is now provided.

4C: Just commenting that NMIIB staining looks exactly as it looks in our laboratory. For some reason, field expects that myosin II would show nice repeating periodicity but it does not. Staining looks a bit scattered as shown here, although there is also periodicity. I think we need to accept this and start to elucidate what does it mean.

We were very pleased to know that similar observations on NMIIB staining were obtained independently.

4G XZ projection and 4J: where is the cell surface and inner "mid-point"? Give directions.

The cell surface and inner mid-point are now indicated in the figure.

5) Figure 3:I cannot see the concluded periodicity for pMLC from figures. To me it looks similar to myosin II heavy chains and my conclusion would be that pMLC and heavy chain are in complex and myosin II complex can be in different orientations. It would be interesting to have pMLC+ heavy chain together in super-resolution. Image might be complicated depending on the sites of labels but at same time, it could give an idea for the orientation of the complex. Would co-imaging of pMLC and myosin heavy chain be possible?

We agree with the reviewer that our data supports that pMLC and NMII can be in complex in axons and that the myosin II complex can be in different orientations. This has been reinforced in the revised version of the manuscript (Subsection “Structural organization and dynamics of actomyosin axonal rings”). Although co-imaging of pMLC and myosin heavy chain was not possible to implement given host antibody constraints, we used an alternative approach to visualize the N- and C-terminus of NMII heavy chains suggested by reviewer 1 (detailed below). Of note, reviewer#1 raised similar concerns. We are transcribing the answer provided:

pMLC is expected to have a periodicity of ~190 nm if myosin is anchored to actin rings. Please note that this periodicity is still compatible with the existence of ~300 nm bipolar NMII filaments in axons (Figure 4O). A 190 nm periodic pMLC staining, coincident with phalloidin, was previously shown restricted to the AIS (Berger et al., 2018). This observation suggested that in the AIS, pMLC is bound to actin rings. Here we show that pMLC periodicity extends to the axon shaft (Figure 3D-F). Interestingly, in some axonal regions, pMLC showed a periodic distribution consistent with anchoring of myosin to different regions of consecutive actin rings (Figure 3D- highlighted by red ruler; and Figure 3F). The absence of a generalized striped pattern for pMLC (as is observed for axonal actin and βII-spectrin staining), supports that in each actin ring a limited number of pMLC molecules is anchored. Alternatively, technical limitations of the antibodies used may preclude the visualization of the entire pool of pMLC molecules present in the structure. Nonetheless, we now provide new imaging data showing that in some regions of the axon shaft, pMLC can be observed with the expected striped pattern, colocalized with actin (Figure 3G). This supports that pMLC is bound to actin rings throughout the axon shaft. This issue is further discussed in the revised version of the manuscript (subsection “Phosphorylated NMII light chains are colocalized with actin within the MPS and organized as circular periodic structures persisting throughout the axon shaft”).

In the case of NMII heavy chains, please note that we imaged it using antibodies that recognize the middle portion of the NMII bipolar filament (Figure 4A-D). Unless NMII heavy chains would be positioned only within individual actin rings, a periodic pattern coincident with phalloidin is not expected. As suggested by the Reviewer, hippocampal neurons and the neuroblastoma cell line SH-SY5Y were transfected with fluorescent NMII fusion constructs that allow visualizing simultaneously the N- (eGFP tag) and C-terminal (mApple tag) of NMII heavy chain (Beach et al., 2014). NMII was then visualized using spinning disk super-resolution microscopy. In the revised version of the manuscript, we show that in axons, NMIIA can assemble into bipolar filaments of ~300 nm in length (Figure 4H and 4I). Similarly, to the data obtained by STORM (Figure 4D), fluorescent NMII fusion proteins revealed the existence of multiple consecutive myosin filaments, with different orientations in relation to the axonal axis. These results are presented and discussed in subsection “NMII heavy chains have multiple orientations in relation to individual actin rings and to the axonal axis, being able to crosslink adjacent rings”.

6) Figure 4K: Model figure: Here are many possibilities. I would ask authors to consider the length of myosin II complexes (bipolar (=300 nm)/ unipolar) and average diameter of an axon ring (here 400 nm, enlarged to 500 nm when contractility was abolished) to get reality to models. As much as I understand, myosin II complex is not changing its length, myosin II heavy chains are not sliding along each other as shown in the model figure.

As requested by the reviewer, the models have been revised to accommodate the regular maximum lengths of approximately 300 nm of NMII filaments, as well as an average axonal diameter of 450 nm.

If d) is the proposed model, I would expect that in xy, myosin II staining should follow the periodicity of actin rings but it does not. To me it seems that there is definitively something between rings. Is it sure that myosin II indeed attach actin filaments in actin rings? Are there other alternative actin filament structures in axons to bind?

This aspect is further discussed in the revised version of the manuscript (subsection “Structural organization and dynamics of actomyosin axonal rings”). The presence of NMII staining that is non-coincident with actin, supports that NMII heavy chains may crosslink different rings. Periodic (~190 nm) pMLC staining, coincident with phalloidin staining restricted to the AIS has been previously shown using STORM (Berger et al., 2018. Thus, binding of pMLC to actin has been hypothesized. Here, as detailed above and further documented in the manuscript, we show that an approximately 190 nm periodicity of pMLC is maintained throughout the axon shaft (Figure 3D-G). Despite this evidence and the fact that NMII activity modulates MPS diameter, the possible interplay between myosinII and deep axonal actin filaments that may also serve as myosin anchors, cannot be ruled out and should certainly be explored in the future. This is further discussed in the revised version of the manuscript (subsection “Structural organization and dynamics of actomyosin axonal rings”).

Reviewer #3:Costa et al. present evidence that myosin II contractility can regulate the diameter of neuronal axons. In addition, they provide striking super resolution imaging that support the proposed orientations of myosin II filaments in relation to the periodic actin rings along axons. This study is of potential interest. However, the authors have failed to provide controls for the vast majority of their experiments, which makes their implications difficult to interpret.1) The authors use hsRNA to knockdown myosin II paralogs. Maybe I somehow missed it but there does not seem to be any evidence presented that the shRNA knockdown the myosin II paralogs in hippocampal neurons. Showing that they work in other cells (i.e., PC-12 or CAD) is not sufficient to make the claim that they are knockdown in primary cultured neurons. The authors should report how much is each myosin II paralog knocked down?

Assays on the efficacy of myosin-II knock-down have been performed in cell lines as the transfection efficiency of hippocampal neurons is very low precluding western blot or qPCR analysis of transfected cells. To address the concern raised by the reviewer, in the revised version of the manuscript we have added to Figure 1—figure supplement 1 the quantification of the efficacy of each shRNA-mediated knockdown in hippocampal neurons, as assessed by quantification of immunofluorescence intensities.

2) The authors make no attempt to show that phenotype reported for each shRNA knockdown is specifically because of the loss of the targeted paralog. The authors should reintroduce full length versions of each targeted paralog to test if it rescues the phenotype and relate the expression amount to endogenous levels in wild type axons. That is, does reintroduction lead to smaller axons?

As suggested by the reviewer, for each of the shRNA experiments, full length shRNA resistant constructs have been used to show rescue. In each case, the use of shRNA resistant constructs led to reversion of phenotypes (Figure 1J,K; subsection “Modulation of NMII activity regulates the expansion and contraction of axonal diameter”).

3) By presented the aforementioned controls for the shRNA knockdowns, the authors will add sufficient support for the use of blebbistatin. However, there is also currently no controls presented to support their claim that ML-7 is inhibiting MLCK in hippocampal neurons. Is the phosphorylation of myosin light chain reduced in the neurons (either mono or di-phosphorylation)? In addition, it would be advisable to test if the ROCK inhibitor, Y-27632, also induces wider axons.

As requested by the reviewer, in the revised version of the manuscript we show using immunofluorescence against pMLC that ML-7 inhibits MLCK in hippocampal neurons (Figure 1—figure supplement 1A and B, subsection “Modulation of NMII activity regulates the expansion and contraction of axonal diameter”).

4) The authors claim that calyculin A is a protein phosphatase 1 (PP1) inhibitor. While this is the case, the omission that it is also a PP2A inhibitor implies a specificity that does not exist. As such, it is advisable to control for the use calyculin A. At the very least, it should be demonstrated that blebbistatin is able to mask the phenotypes induced by calyculin A.

As suggested by the reviewer, we now show that blebbistatin reverts the phenotype induced by calyculin A (Figure 1D,E; subsection “Modulation of NMII activity regulates the expansion and contraction of axonal diameter”).

5) What is the evidence that myosin V is being inhibited in the neurons by Myovin-1?

Myovin-1 specifically arrests the myosinV ATP/ADP cycle, thus inhibiting myosin V as a processive motor (Islam et al., 2010). Recent studies have further used myovin-1 in hippocampal neurons to inhibit myosinV-dependent intersynaptic vesicle exchange in live cells (Gramlich et al., 2017). This has been further clarified in the text, subsection “Modulation of NMII activity regulates the expansion and contraction of axonal diameter”.

6) It does not seem to me that the singular use of blebbistatin is enough to make any claims as to the role of myosin II in regulating propagation velocity. This data seems preliminary at best.

In the current version of the manuscript, we further explored the effect of blebbistatin in the regulation of propagation velocity, by increasing the number of analyses performed and by measuring the mean firing rate (MFR) in both blebbistatin-treated and untreated neurons. Also, the need to further explore this effect through the use of additional modulators of NMII activity and shRNA-mediated downregulation approaches, is now discussed (subsection “Propagation velocity is altered by manipulation of NMII activity”).